# PVDepth: Panoramic Video Depth Estimation via Geometry-Aware Spatiotemporal Adaptation

**Chuanxin Song** [1 2]   **Peixi Peng** [1 2]

## Abstract

Panoramic video depth estimation is pivotal for applications such as Virtual Reality and World Models. However, advancements in this field are impeded by two primary obstacles: the scarcity of large-scale training data and the unique spatiotemporal challenges of Equirectangular Projection (ERP), which hinder the direct transfer of perspective models. In this paper, we first present **PanoCARLA**, a large-scale synthetic RGB-D panoramic video dataset, featuring natural motion trajectories and drone-like roaming perspectives. Building on this foundation, we propose **PVDepth**, an end-to-end framework adapted from perspective video depth models. To tackle ERP-specific geometric distortions and consequent non-linear temporal dynamics, we introduce two core mechanisms: (1) A *Progressive Sphere-aware Noise Initialization* strategy that anneals the noise distribution from planar to spherical, guiding the model to adapt to non-uniform information density; and (2) A *Cube-rectified Temporal Modeling* module that incorporates an auxiliary cubemap temporal branch to rectify non-linear temporal dynamics in the ERP domain. Extensive experiments demonstrate that PVDepth achieves superior performance, generating geometrically accurate and temporally consistent depth sequences. Code and data will be released at https://github.com/ChuanxinSong/PVDepth.

## 1. Introduction

Panoramic vision plays a vital role in modern applications (Ai et al., 2025), ranging from virtual reality (Zhou et al., 2024) to robot navigation (Zheng et al., 2025; Gat-

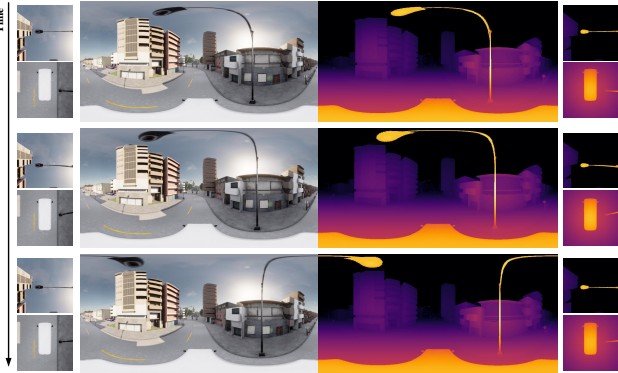

*Figure 1.* Illustration of the unique spatiotemporal challenges within ERP. For each timestamp, the side insets display the distortion-free up and down cubemap views. *Spatially*, ERP severely stretches the polar regions, allocating excessive pixels to represent sparse information. *Temporally*, ERP's geometric distortion causes smooth object motion to manifest as discontinuous, non-linear dynamics in the polar regions.

taux et al., 2025) and world models (Team et al., 2025). A critical step in these domains is recovering temporally consistent depth, which transforms 2D video sequences into 3D space. This geometric understanding is essential for enabling immersive interaction, motivating our focus on panoramic video depth estimation.

Although significant progress has been made in single-image panoramic depth estimation (Li et al., 2025a), as shown in Fig. 4, the lack of temporal constraints often results in severe temporal inconsistency when applied directly to video sequences. While post-processing alignment methods exist, they are usually computationally inefficient and subject to strict constraints: either assuming a static viewpoint that fails to support camera movement (Zhou et al., 2025) or struggling to handle dynamic objects with independent motion trajectories (Yang et al., 2025b). More critically, the scarcity of large-scale panoramic RGB-D video datasets has hindered the emergence of end-to-end panoramic video depth estimation frameworks.

To overcome the aforementioned data bottleneck, we construct **PanoCARLA**, a large-scale panoramic RGB-D video

[1]School of Electronic and Computer Engineering, Peking University, Shenzhen, China [2]Pengcheng Laboratory, Shenzhen, China. Correspondence to: Peixi Peng <pxpeng@pku.edu.cn>.

*Proceedings of the 43rd International Conference on Machine Learning*, Seoul, South Korea. PMLR 306, 2026. Copyright 2026 by the author(s).

dataset collected in the CARLA simulator (Dosovitskiy et al., 2017), capturing sequences along both natural vehicle trajectories and drone-like roaming viewpoints. With the PanoCARLA dataset, an intuitive idea is to directly fine-tune SOTA perspective video depth models (Hu et al., 2025) on panoramic sequences. However, mere fine-tuning is insufficient, due to the unique geometric properties of Equirectangular Projection (ERP).

*Spatially*, models pretrained on perspective imagery inherit inductive biases from distortion-free projections, where pixels correspond to approximately uniform sampling. However, as shown in Fig. 1, ERP introduces severe latitude-dependent stretching, particularly in the polar regions. As a result, the polar regions are heavily over-sampled in the pixel domain despite containing limited effective information. This sampling mismatch distracts the model from learning valid structural features and may lead to overfitting to geometric distortions. *Temporally*, as shown in Fig. 1, ERP's geometric distortion causes object motion to manifest as highly non-linear pixel trajectories in the polar regions. Standard video depth backbones (Chen et al., 2025; Yang et al., 2025a; Hu et al., 2025) typically apply 1D attention solely along the temporal dimension ($T$) of the feature tensor $(B, H, W, T, C)$. While computationally efficient, this design is primarily effective when local motion is approximately linear. Consequently, it struggles to capture temporal correlations from such irregular motion patterns.

To address these challenges, we propose **PVDepth**, an end-to-end panoramic video depth estimation framework, building upon a pretrained perspective-domain model (Hu et al., 2025) via spatiotemporal adaptation. First, to bridge the geometric mismatch between planar priors and the ERP's spherical topology, we introduce a *Progressive Sphere-aware Noise Initialization* strategy. Utilizing an annealed weighting mechanism, we smoothly transition the noise initialization from the planar to the spherical domain. This curriculum-style adaptation explicitly guides the model to progressively adapt to the non-uniform information density in ERP. Second, to handle the non-linear temporal dynamics, we design a *Cube-rectified Temporal Modeling* module. Observing that standard temporal layers struggle with non-linear motion patterns in ERP, we introduce an auxiliary branch that performs temporal modeling in the distortion-free cube representation. The resulting Cube features are then mapped back to the ERP domain to rectify ERP-induced non-linear temporal dynamics in the primary branch. Our key contributions are:

- We introduce **PanoCARLA**, a large-scale panoramic RGB-D video dataset designed to alleviate the data scarcity in panoramic video depth estimation.

- We propose **PVDepth**, an end-to-end panoramic video depth estimation framework. It incorporates a *Pro-*

*gressive Sphere-aware Noise Initialization* and a *Cube-rectified Temporal Modeling* module to tackle the unique spatiotemporal challenges of ERP.

- Extensive experiments demonstrate that PVDepth consistently outperforms existing methods, producing depth sequences with improved geometric accuracy and temporal consistency.

## 2. Related Works

**Panoramic Depth Estimation.** Monocular panoramic depth estimation remains a challenging task due to the severe geometric distortions inherent in ERP. Several approaches aim to rectify distortions through specialized architectures, either by fusing features from low-distortion projections (e.g., Cubemaps) (Wang et al., 2022; Ai & Wang, 2024) or by introducing spherical geometry priors into Transformers (Shen et al., 2022; Benny & Wolf, 2025). Considering the scarcity of panoramic data, another paradigm involves transferring rich priors from the perspective domain to facilitate panoramic estimation. Representative strategies include decomposing panoramas into perspective patches for separate estimation (Rey-Area et al., 2022; Jung et al., 2025), or distilling robust perspective abilities (Yang et al., 2024) into panoramic models via semi-supervised learning (Wang & Liu, 2024; Cao et al., 2025) or generative data curation (Li et al., 2025a). Despite the remarkable progress, these methods are fundamentally tailored for single-image estimation without cross-frame temporal consistency; direct application to video sequences often results in flickering (Hu et al., 2025). In contrast, our work aims to generate temporally consistent depth for panoramic videos.

**Video Depth Estimation.** Video depth estimation has witnessed remarkable progress, propelled by the emergence of powerful foundation models (Oquab et al., 2023) and generative priors (Blattmann et al., 2023). Specifically, Video Depth Anything (Chen et al., 2025) utilizes spatial-temporal heads to enforce gradient consistency, while generative approaches like DepthCrafter (Hu et al., 2025) and Depth Any Video (Yang et al., 2025a) harness the rich priors of video diffusion models to synthesize high-fidelity depth sequences. Even so, directly applying these models to panoramic video remains non-trivial due to the domain gap caused by ERP distortions. Existing panoramic works (Huang et al., 2025; Zhou et al., 2025; Yang et al., 2025b) rely on a "predict-then-optimize" paradigm. They typically estimate per-frame depth followed by iterative geometric alignment to enforce consistency. Despite being effective, such optimization is computationally expensive and struggles with complex dynamic scenes. In contrast, we propose an end-to-end framework, PVDepth, which achieves temporally consistent depth estimation without any test-time optimization.

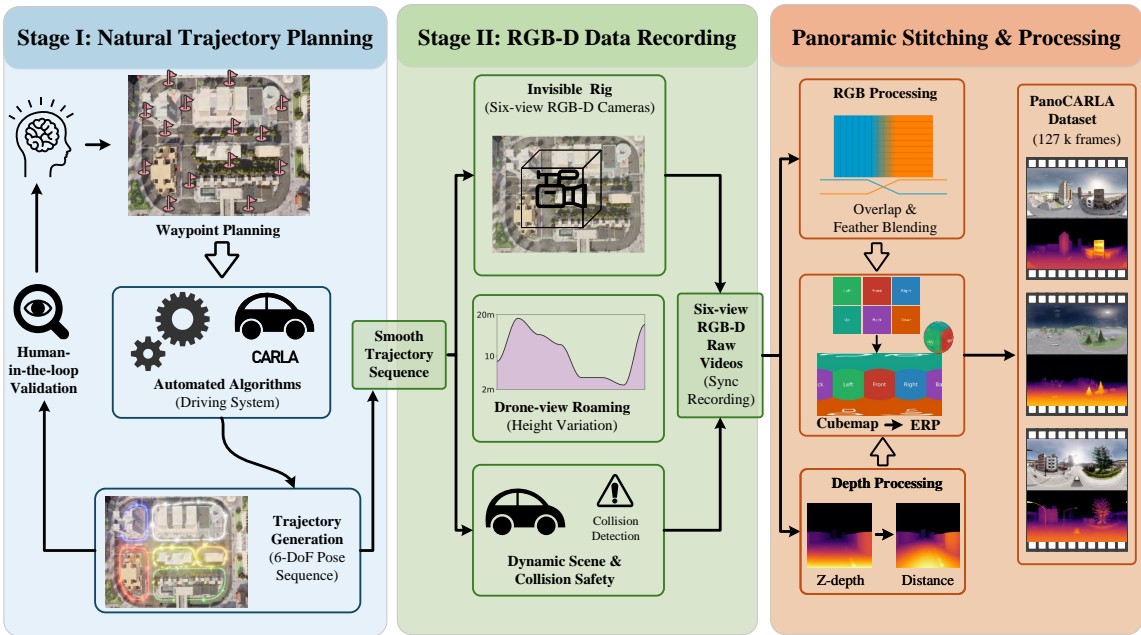

*Figure 2.* Overview of the **PanoCARLA** construction pipeline. In *Stage I*, human-in-the-loop waypoint adjustment ensures smooth and realistic 6-DoF trajectories. In *Stage II*, an unobstructed recording setup enables drone-view roaming with dynamic altitude variations, supported by six-view collision detection to eliminate environment-clipping artifacts. *Finally*, the collected data are processed modality-specifically, yielding seamlessly stitched RGB panoramas and geometrically consistent depth annotations.

**Synthetic Datasets for Depth Estimation.** While real-world datasets are fundamental to depth estimation, they often suffer from sensor noise and sparse annotations. Recent advancements, such as Depth Anything V2 (Yang et al., 2024) and Marigold (Ke et al., 2024), demonstrate that high-quality synthetic data offers pixel-perfect supervision and significantly enhances Sim-to-Real generalization. This advantage extends to the video domain, where Depth Any Video (Yang et al., 2025a) and DKT (Xu et al., 2025) leverage large-scale synthetic sequences to overcome the difficulty of capturing temporally aligned ground truth in dynamic real-world environments. In panoramic vision, synthetic datasets such as Structured3D (Zheng et al., 2020) and Realsee3D (Li et al., 2025b) have facilitated static perception tasks. However, current datasets remain confined to static environments. The lack of temporal dimension restricts the development of video-specific approaches. To bridge this gap, we introduce a large-scale panoramic RGB-D video dataset, PanoCARLA, to extend the advantages of synthetic data to panoramic video depth estimation.

## 3. PanoCARLA Dataset

To address the scarcity of RGB-D panoramic video data, we introduce **PanoCARLA**, a large-scale synthetic dataset curated within CARLA simulator (Dosovitskiy et al., 2017). While existing approaches often synthesize trajectories by

*Table 1.* **Comparison with representative panoramic RGB-D datasets.**

| Dataset | Scene | Depth | Syn./Real | Video | #Frames |
|---|---|---|---|---|---|
| Structured3D (Zheng et al., 2020) | Indoor | ✓ | Syn. | ✗ | 196K |
| Matterport3D (Chang et al., 2017) | Indoor | ✓ | Real | ✗ | 10K |
| Stanford2D3D (Armeni et al., 2017) | Indoor | ✓ | Real | ✗ | 1K |
| Omni360-Scene (Ge et al., 2025) | Outdoor | ✓ | Syn. | ✗ | 61K |
| **PanoCARLA** | **Outdoor** | ✓ | **Syn.** | ✓ | **126.7K** |

interpolating between discrete waypoints (Lu et al., 2024), we capture more natural motion by mounting cameras directly onto autonomous vehicles to leverage the simulator's built-in navigation policies.

However, this setup introduces a critical issue: the vehicle body inevitably causes persistent self-occlusion (see Fig. 5). To resolve this, we design a *Two-Stage Decoupled Data Acquisition Pipeline*, as illustrated in Fig. 2. The pipeline separates trajectory acquisition from scene rendering:

**Stage I: Natural Trajectory Planning.** We generate smooth driving paths by designating sparse waypoints and leveraging CARLA's Autopilot for physically consistent navigation. We record the resulting 6-DoF vehicle poses at 20 FPS, incorporating human-in-the-loop refinement to iteratively adjust waypoints for optimal smoothness.

**Stage II: Unobstructed RGB-D Data Recording.** We replay the recorded poses to drive a "ghost rig"—a virtual

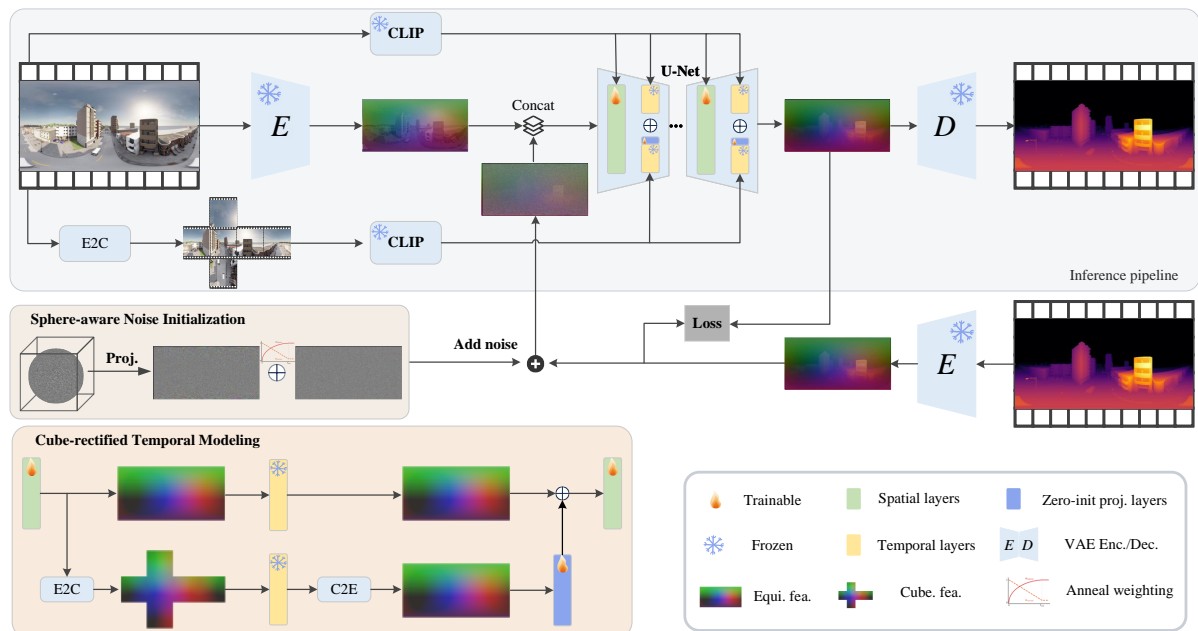

*Figure 3.* Overview of the proposed **PVDepth**. Building upon a pre-trained video diffusion backbone, PVDepth incorporates two key spatiotemporal adaptations to bridge the geometric gap between perspective and panoramic domains: (1) *Progressive Sphere-aware Noise Initialization*, which smoothly transitions the input noise from a planar to a spherical distribution to adapt to non-uniform ERP information density; and (2) *Cube-rectified Temporal Modeling*, which introduces an auxiliary distortion-free Cubemap branch to capture linear motion modeling, rectifying ERP-induced non-linear dynamics in the primary ERP branch.

assembly of six synchronized RGB-D cameras detached from vehicles. This setup preserves realistic motion while providing a 360° unobstructed view. To further enhance viewpoint diversity, we introduce Drone-view Roaming, where the camera altitude is smoothly varied (2m–20m) to simulate 6-DoF aerial motion. In addition, a six-view collision detector is employed to identify frames where the camera intersects with environmental structures, enabling the automatic removal of frames with clipping artifacts.

**Panoramic Stitching & Processing.** Since CARLA lacks panoramic sensors, we capture six-view cubemaps and stitch them into the ERP format with modality-specific optimizations. *For RGB*, we utilize a 120° Field of View (FoV) for each camera to ensure sufficient overlap. Applying *feather blending* in overlap regions effectively mitigates potential seams caused by exposure discrepancies across cameras. *For depth maps*, we perform a pixel-wise calibration to ensure geometric consistency. Since CARLA's raw depth output is planar Z-depth, which is incompatible with the spherical geometry of an ERP, we convert it into Euclidean distance before stitching. This yields seam-free geometry, so depth stitching does not involve overlapping regions and is performed by direct composition.

**Dataset Statistics.** PanoCARLA encompasses diverse urban environments (Town01–Town07, Town10) and features

dynamic traffic. In total, it comprises 117 long video clips with **126.7K frames**, covering over 50 km of roaming distance. For each frame, we provide RGB-D panoramas and camera extrinsics. We further compare PanoCARLA with representative panoramic RGB-D datasets in Table 1. Unlike prior datasets that primarily focus on indoor scenes or static imagery, PanoCARLA provides large-scale *outdoor RGB-D panoramic videos*. More dataset statistics and details are provided in Appendix A.

## 4. Methodology

### 4.1. Overview of PVDepth

Recent research has highlighted the potential of generative priors for depth estimation. While Marigold (Ke et al., 2024) pioneered this paradigm for images, DepthCrafter (Hu et al., 2025) successfully extended it to videos. We adopt DepthCrafter as our baseline to explore extending the generative paradigm to panoramic video depth estimation.

As shown in Fig. 3, the input RGB video $v$ is encoded into the video latent $\mathbf{z}^{(v)} = \mathcal{E}(\mathbf{v})$ via a pre-trained VAE. For the ground-truth depth $d$, we convert it into disparity and replicate the single-channel map three times to match the VAE's input dimensionality, obtaining $\mathbf{z}^{(d)} = \mathcal{E}(\mathbf{d})$. Following perspective approaches (Hu et al., 2025), we predict relative

(affine-invariant) depth, normalized to the range $[0, 1]$. During inference, the denoising U-Net takes the noisy depth latent $z_t^{(d)}$ as input. To enforce pixel-aligned correspondence, the clean video latent $z^{(v)}$ is frame-wise concatenated with $z_t^{(d)}$ along the channel dimension. Semantic guidance is injected via frame-level CLIP embeddings through cross-attention layers. The VAE then decodes the denoised latent, and the three output channels are averaged to produce the final single-channel depth sequence. See the detailed diffusion formulation in Appendix B.

While effective for perspective video, directly applying DepthCrafter to panoramic video suffers from ERP's geometric distortions. To bridge this geometric domain gap, we extend the baseline with geometry-aware spatiotemporal adaptation for panoramic videos. This is realized through two core designs: Progressive Sphere-aware Noise Initialization and Cube-rectified Temporal Modeling.

## 4.2. Progressive Sphere-aware Noise Initialization

Standard video diffusion models (Hu et al., 2025; Blattmann et al., 2023) assume the initial noise follows an i.i.d. Gaussian distribution, $\epsilon \sim \mathcal{N}(\mathbf{0}, \mathbf{I})$. While suitable for perspective images with approximately uniform information density, this assumption becomes suboptimal for ERP, where the sphere-to-ERP mapping induces a highly non-uniform information distribution (see Fig. 7). Since diffusion models rely on noise to explore and denoise the data manifold, spatially uniform noise in ERP implicitly guides the model toward over-represented but low-information regions, hindering its adaptation to the underlying spherical geometry.

To explicitly guide the model to adapt to the non-uniform information density in ERP, inspired by SpND (Sun et al., 2025), we reformulate the noise initialization as *Sphere-aware Noise*. As illustrated in Fig. 3, we sample Gaussian noise in a geometry-consistent 3D space and project it onto the ERP representation: $\epsilon_s = \mathcal{P}_{S \rightarrow E}(\epsilon_{sphere})$. This formulation is consistent with how ERP images are obtained in practice, where a spherical signal is unfolded into the equirectangular projection. As a result, the induced ERP noise is representation-wise consistent with the underlying spherical geometry, with its information density naturally aligned with the sphere-to-ERP projection.

Distinct from SpND which relies on an auxiliary branch to inject geometric priors, our approach directly modifies the noise distribution, thereby avoiding additional parameter burden. However, immediately enforcing this spherical distribution can lead to suboptimal convergence, as the sudden shift from the pre-trained planar distribution destabilizes training. To address this issue, we introduce a Progressive Adaptation Strategy (Song et al., 2024), in which the initial

noise is formulated as a variance-preserving mixture:

$$\epsilon = w_s \cdot \epsilon_s + w_n \cdot \epsilon_n, \quad \text{s.t.} \quad w_s = \sqrt{1 - w_n^2}, \quad (1)$$

where $\epsilon$ denotes the final noise injected into the network, composed of the normal planar noise $\epsilon_n$ and the sphere-aware noise $\epsilon_s$ with weights $w_n$ and $w_s$. The planar weight $w_n$ linearly decays from 1 to a small residual value $w_{final}$ during the early training stages, explicitly ensuring that the variance of $\epsilon$ remains equal to 1. This strategy allows the model to leverage robust planar pre-trained priors at early stages, while progressively adapting to the non-uniform spherical geometry. Empirically, we find that retaining a minimal residual of planar noise (e.g., $w_{final} = 0.1$) yields better performance than completely removing it. More details and pseudocode are provided in Appendix C.

This geometry-aware noise initialization introduces no additional learnable parameters and incurs negligible computational overhead. By aligning the injected noise with the sphere-to-ERP sampling density, it enables efficient adaptation to panoramic geometry without modifying the denoising architecture.

## 4.3. Cube-rectified Temporal Modeling

Standard video depth backbones (Hu et al., 2025; Chen et al., 2025) rely on 1D temporal modules to enforce consistency across frames. Given a feature tensor $\mathbf{F} \in \mathbb{R}^{B \times T \times C \times H \times W}$, these modules typically compute attention solely along the temporal dimension $T$, implicitly assuming that local motion trajectories are approximately linear within the receptive field. However, this assumption does not hold in ERP, where latitude-dependent stretching induces highly non-linear dynamics in polar regions (see Fig. 1). Consequently, applying standard 1D temporal attention directly to ERP features struggles to capture these irregular motion dynamics, leading to suboptimal depth results.

As shown in Fig. 1, Cubemap projection provides a distortion-free representation of polar regions. Consequently, the temporal dynamics within Cube domain exhibit near-linear properties, which aligns well with 1D temporal layers. Inspired by this observation, as shown in Fig. 3, we propose the *Cube-rectified Temporal Modeling* module, leveraging Cube-aware temporal features to assist the temporal modeling of ERP features.

Specifically, we introduce an auxiliary Cubemap branch alongside the primary ERP branch. Let $\mathbf{z}_{erp} \in \mathbb{R}^{B \times T \times C \times H \times W}$ denote the latent features from spatial layers and $\mathcal{T}(\cdot, \mathbf{c})$ denote the temporal layer conditioned on context $\mathbf{c}$. In the auxiliary branch, we first project the ERP features into the Cubemap representation $\mathbf{z}_{cube} \in \mathbb{R}^{(B \times 6) \times T \times C \times H_c \times W_c}$ (where the batch dimension is expanded by the 6 faces), and then apply the temporal layer to

*Table 2.* **Quantitative comparison.** The **best** and underline{second} best performances are **bolded** and underlined. ZS and FT denote Zero-Shot and Fine-Tuned settings. (Cube-based) refers to the strategy of inferring on decomposed cubemap faces independently and stitching them back. Gray values denote frame-level performance of image-based methods.

| Method | fps02_len50 | | fps10_len90 | | fps20_len110 | |
|---|---|---|---|---|---|---|
| | AbsRel $\downarrow$ | $\delta_1 \uparrow$ | AbsRel $\downarrow$ | $\delta_1 \uparrow$ | AbsRel $\downarrow$ | $\delta_1 \uparrow$ |
| *Panorama image depth-based* | | | | | | |
| DA-2$_{ZS}$ (Li et al., 2025a) | 1.367 (0.781) | 0.573 (0.742) | 1.125 (0.759) | 0.617 (0.759) | 1.021 (0.731) | 0.656 (0.761) |
| UniK3D$_{ZS}$ (Piccinelli et al., 2025) | 0.644 (0.486) | 0.610 (0.724) | 0.580 (0.455) | 0.638 (0.741) | 0.551 (0.442) | 0.666 (0.742) |
| PanoDA$_{ZS}$ (Cao et al., 2025) | 0.767 (0.527) | 0.262 (0.344) | 0.721 (0.526) | 0.281 (0.342) | 0.642 (0.524) | 0.298 (0.341) |
| DepthAnywhere$_{ZS}$ (Wang & Liu, 2024) | 0.487 (0.260) | 0.405 (0.788) | 0.493 (0.220) | 0.444 (0.806) | 0.368 (0.220) | 0.543 (0.807) |
| DepthAnywhere$_{FT}$ (Wang & Liu, 2024) | 0.469 (**0.214**) | 0.421 (**0.833**) | 0.506 (**0.204**) | 0.455 (**0.853**) | 0.400 (**0.204**) | 0.572 (**0.852**) |
| *Post-processing alignment-based* | | | | | | |
| Matrix-3D (Yang et al., 2025b) | 0.601 | 0.386 | 0.512 | 0.512 | 0.369 | 0.640 |
| ViPE (UniK3D-based) (Huang et al., 2025) | 0.684 | 0.701 | 0.679 | 0.730 | 0.648 | 0.731 |
| *Perspective video depth-based* | | | | | | |
| VideoDepthAnything$_{ZS}$ (Chen et al., 2025) | 0.403 | 0.459 | 0.359 | 0.488 | 0.306 | 0.569 |
| DepthAnyVideo$_{ZS}$ (Yang et al., 2025a) | 0.315 | 0.562 | 0.318 | 0.595 | 0.252 | 0.667 |
| DepthCrafter(Cube-based)$_{ZS}$ | 1.480 | 0.257 | 0.712 | 0.337 | 0.630 | 0.390 |
| DepthCrafter(Cube-based)$_{FT}$ | 0.543 | 0.547 | 0.395 | 0.589 | 0.313 | 0.635 |
| DepthCrafter$_{ZS}$ (Hu et al., 2025) | 0.340 | 0.498 | 0.287 | 0.563 | 0.244 | 0.640 |
| DepthCrafter$_{FT}$ (Hu et al., 2025) | 0.228 | 0.752 | 0.252 | 0.729 | 0.216 | 0.781 |
| **PVDepth (Ours)** | **0.211** | **0.789** | **0.205** | **0.787** | **0.181** | **0.818** |

process these linearized features:

$$\mathbf{z}_{cube} = \mathcal{P}_{E \to C}(\mathbf{z}_{erp}), \quad \mathbf{z}'_{cube} = \mathcal{T}(\mathbf{z}_{cube}, \mathbf{c}_{cube}), \quad (2)$$

where $\mathcal{T}$ shares weights with the primary branch to maximize parameter efficiency, and $\mathbf{c}_{cube}$ is derived from the corresponding Cubemap images to explicitly ensure semantic alignment.

After modeling the temporal dynamics in linearized Cube features, we project them back to the ERP format: $\mathbf{z}_{rec} = \mathcal{P}_{C \to E}(\mathbf{z}'_{cube})$. This correction signal is then fused into the primary ERP branch via a zero-initialized projection (Zhang et al., 2023) for stability:

$$\mathbf{z}_{out} = \mathcal{T}(\mathbf{z}_{erp}, \mathbf{c}_{erp}) + \mathcal{Z}(\mathbf{z}_{rec}). \quad (3)$$

This strategy allows the model to progressively incorporate cube features to rectify non-linear polar dynamics.

## 5. Experiments

### 5.1. Experimental Settings

**Dataset Setup.** Given the absence of publicly available RGB-D panoramic video datasets, we conduct experiments on our PanoCARLA dataset. We adopt a strict cross-scene evaluation protocol to evaluate generalization to unseen environments. Specifically, the model is trained on sequences from Towns 01, 03–07 (∼113k frames) and evaluated on a held-out split comprising Towns 02 and 10 (∼14k frames). From these held-out sequences, we curate a test set of 197 video clips. To comprehensively evaluate adaptability to diverse temporal dynamics, these clips are categorized into

three settings: (1) *Low FPS*: 2 fps (50 frames) with 17 clips; (2) *Medium FPS*: 10 fps (90 frames) with 66 clips; and (3) *High FPS*: 20 fps (110 frames) with 114 clips.

In addition, we conduct qualitative comparisons on open-world panoramic videos from the Web360 dataset (Wang et al., 2024), where no depth annotations are available. Further, to quantitatively evaluate performance on real-world data without depth supervision, we introduce *a reprojection-based geometric consistency evaluation protocol,* with detailed settings and results provided in Appendix F.

**Implementation Details.** PVDepth is initialized from DepthCrafter (Hu et al., 2025) and trained in two stages. In Stage I, we fine-tune spatial layers for 20K steps with PSNI to adapt the model to ERP geometry. In Stage II, we introduce the CRTM module and fine-tune only the newly added projection layers for another 20K steps. We use AdamW with a learning rate of $5 \times 10^{-6}$ and a batch size of 8, and randomly sample frame rates from $[1, 20]$ during training to cover diverse motion dynamics. Following DepthCrafter, we train at $640 \times 320$ and use a 5-step denoising schedule with sliding-window inference for arbitrary-length videos. Evaluation is conducted at $1024 \times 512$, a commonly used panoramic video resolution (Wang et al., 2024). More implementation details are provided in Appendix D.

**Evaluation Metrics.** We report standard metrics including AbsRel and $\delta_1$. Following video depth benchmarks (Chen et al., 2025; Hu et al., 2025), we perform affine-invariant evaluation using *Global Alignment*, where a single scale and shift are optimized for the entire video clip to faithfully reflect temporal consistency. Detailed metric definitions and alignment protocols are provided in Appendix E.

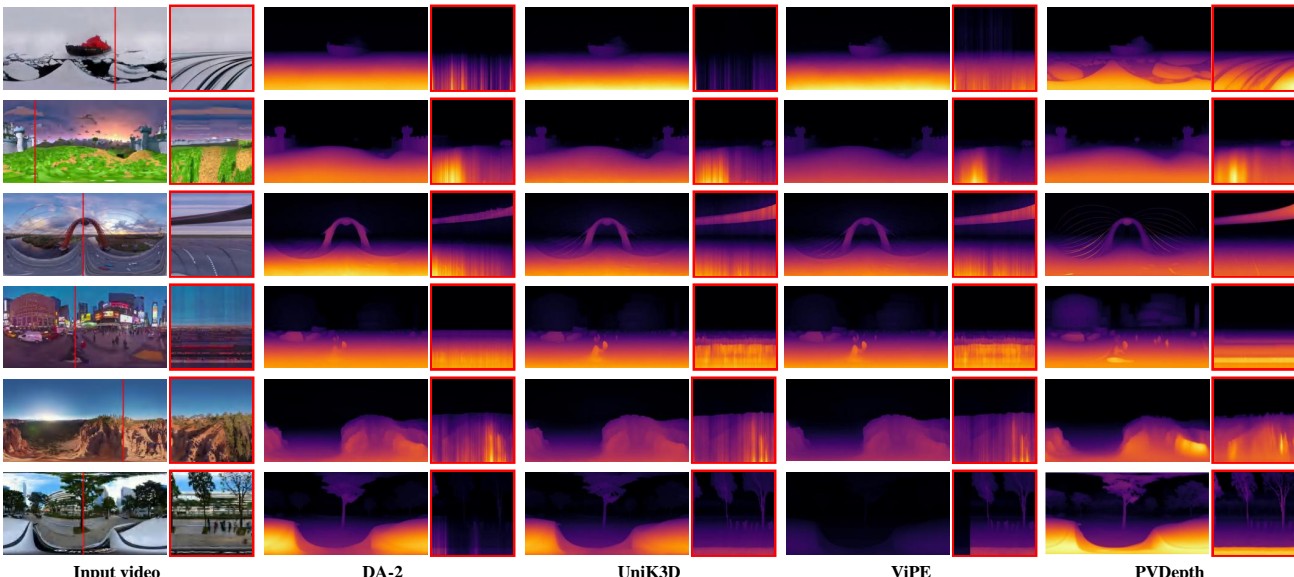

*Figure 4.* **Qualitative comparisons on in-the-wild panoramic videos.** To visualize the temporal consistency, we show the temporal profiles in red boxes, obtained by slicing the depth values along the time axis at the vertical red line positions.

## 5.2. Comparison with State-of-the-Arts

**Quantitative results.** We compare PVDepth with three categories of methods, as summarized in Table 2. Detailed configurations are provided in Appendix G.

*1) Image-based methods.* Although image-based panoramic depth methods achieve strong per-frame performance (indicated by gray values), they perform poorly under the video evaluation setting due to the lack of temporal constraints. We further fully fine-tuned DepthAnywhere (Wang & Liu, 2024) on PanoCARLA using its official training code. While fine-tuning improves its single-frame performance, its video performance remains limited due to inconsistent scales and flickering across frames.

*2) Post-processing methods.* Methods that enforce temporal consistency through post-processing also show limitations. Matrix-3D (Yang et al., 2025b) relies on reprojection consistency, limiting it to static environments and resulting in poor performance on our dynamic dataset. ViPE (Huang et al., 2025), an iterative optimization based on UniK3D (Piccinelli et al., 2025), improves $\delta_1$ but degrades AbsRel compared to its baseline. This may be related to ViPE's semantic masking strategy: its global affine depth alignment is estimated primarily from static-background regions, so the resulting correction may not transfer equally well to moving foreground objects in dynamic scenes.

*3) Perspective video methods.* Perspective video depth methods exhibit limited zero-shot performance on panoramic data due to the significant domain gap. Even attempting to decompose the panorama into Cubemaps for infer-

ence destroys the global context (see Fig. 10), yielding results worse than direct ERP input. In contrast, fine-tuning DepthCrafter (Hu et al., 2025) on PanoCARLA yields a substantial performance boost, demonstrating the necessity of training with panoramic data. Furthermore, our PVDepth achieves additional improvements over this strong baseline, further demonstrating the effectiveness of our proposed spatiotemporal adaptation modules.

**Qualitative results.** To further assess real-world generalization in the absence of depth ground truth, we present qualitative comparisons on open-world panoramic videos from the Web360 dataset (Wang et al., 2024). Following (Hu et al., 2025), as shown in Fig. 4, we visualize temporal consistency by extracting temporal profiles along the vertical scanlines (marked in red) in videos. It can be observed that single-frame methods (DA-2 (Li et al., 2025a) and UniK3D (Piccinelli et al., 2025)) exhibit distinct zigzag artifacts along the time dimension, indicating severe interframe flickering due to independent per-frame prediction. Although ViPE (Huang et al., 2025) alleviates some jittering through iterative alignment, it still suffers from residual instability. In contrast, PVDepth generates smooth transitions in the temporal slices, demonstrating superior generalization and temporal consistency in open-world scenarios. We also provide frame-wise qualitative visualizations in Appendix H to complement the temporal profile analysis.

### 5.3. Ablation Studies

In this section, we conduct comprehensive ablation studies to validate the effectiveness of individual components

in PVDepth, specifically the **P**rogressive **S**phere-aware **N**oise **I**nitialization (**PSNI**) and **C**ube-**r**ectified **T**emporal **M**odeling (**CRTM**). For brevity and clarity, we report the averaged performance across the three evaluation settings defined in Sec. 5.1.

**Effectiveness of Proposed Components.** We first evaluate the individual contributions of our core modules in Table 3. The primary results report models trained for 20k steps. To ensure a fair comparison with the total computational budget of our two-stage PVDepth, we also provide results trained for 40k steps (in parentheses). It is observed that simply extending the training duration for the baseline yields marginal gains or even degradation due to overfitting. In contrast, applying either PSNI or CRTM independently yields clear performance gains. Notably, the full PVDepth integrates both modules to achieve superior performance. These results demonstrate that our spatial (PSNI) and temporal (CRTM) adaptations are complementary, collectively addressing the unique geometric challenges of ERP.

**Analysis of Noise Initialization Strategy.** In Table 4, we investigate the optimal design for noise initialization. While directly employing sphere-projected noise (*Spherical*) improves over the baseline, the gain is limited. This is likely because the abrupt noise distribution shift hinders the optimal utilization of pre-trained priors. Our annealing strategy (*Anneal*) bridges this gap, allowing the model to smoothly transition from the perspective to spherical domain in a curriculum learning manner. Furthermore, we observe that retaining a small residual of normal noise (*Res. Normal*) yields the optimal performance. This implies that preserving a component of the original noise distribution may facilitate better alignment with the pre-trained priors.

**Design of Cube-rectified Temporal Modeling.** In Table 5, we investigate the design of CRTM module. First, compared to our learnable fusion strategy ("*Proj. & Add*"), naive fusion methods (e.g., direct addition or averaging) lead to performance degradation. This demonstrates the necessity of the projection layer for stable integration. Second, regarding the attention mechanism, using corresponding Cube semantic features as the conditioning context (*Key/Value source*) yields better results compared to using Equi features. This confirms that leveraging spatially aligned semantic guidance facilitates more effective temporal modeling.

**Exploration of Training Strategy.** Table 6 investigates the training protocol. First, the *One-Stage Joint* strategy yields suboptimal convergence, likely due to optimization interference between the anneal geometric adaptation via PSNI and temporal alignment via CRTM. In contrast, the *Two-Stage Protocol* improves performance by decoupling these two optimization processes. Specifically, freezing the spatial layers in the second stage ("*Two-Stage Frozen*") outperforms keeping them trainable ("*Two-Stage Unfrozen*").

*Table 3.* **Effectiveness of Proposed Components.** *Values in parentheses* denote results trained for 40k steps to match the total computational budget of the full model.

| Method | Components | | Metrics | |
|---|---|---|---|---|
| | *PSNI* | *CRTM* | AbsRel $\downarrow$ | $\delta_1 \uparrow$ |
| Baseline | - | - | 0.248 (0.232) | 0.757 (0.754) |
| *w/ PSNI* | ✓ | - | 0.228 (0.215) | 0.782 (0.784) |
| *w/ CRTM* | - | ✓ | 0.208 (0.213) | 0.779 (0.777) |
| **PVDepth (Full)** | ✓ | ✓ | **0.199** | **0.798** |

*Table 4.* **Ablation studies on Noise Initialization Strategy.** *Spherical*: Sphere-projected noise; *Anneal*: Progressive transition strategy; *Res. Normal*: Residual normal Gaussian noise.

| Noise Strategy | | | CRTM | Metrics | |
|---|---|---|---|---|---|
| *Spherical* | *Anneal* | *Res. Normal* | | AbsRel $\downarrow$ | $\delta_1 \uparrow$ |
| - | - | - | - | 0.248 | 0.757 |
| ✓ | - | - | - | 0.235 | 0.758 |
| ✓ | - | ✓ | - | 0.231 | 0.770 |
| ✓ | ✓ | - | - | 0.228 | 0.782 |
| ✓ | ✓ | ✓ | - | 0.229 | 0.785 |
| ✓ | ✓ | - | ✓ | 0.203 | 0.795 |
| ✓ | ✓ | ✓ | ✓ | **0.199** | **0.798** |

*Table 5.* **Ablation on CRTM Design.** KV Source: Source for Attention keys/values in the temporal layers.

| Method Variant | Fusion Strategy | KV Source | Metrics | |
|---|---|---|---|---|
| | | | AbsRel $\downarrow$ | $\delta_1 \uparrow$ |
| Equi-only (Baseline) | - | Equi | 0.248 | 0.757 |
| Naive Average | Avg | Equi + Equi | 0.267 | 0.704 |
| Naive Add | Add | Equi + Equi | 0.356 | 0.649 |
| CRTM (Self-guided) | Proj. & Add | Equi + Equi | 0.214 | 0.772 |
| **CRTM (Cube-guided)** | **Proj. & Add** | **Equi + Cube** | **0.208** | **0.779** |

*Table 6.* **Ablation studies on Training Strategy.** We compare *One-Stage Joint* (simultaneously optimizing spatial and temporal modules) against a *Two-Stage Protocol* ($1_{th}$: Spatial Adaptation via PSNI; $2_{th}$: Temporal Modeling via CRTM). *Spatial Update*: Whether spatial weights are updated during $2_{th}$ stage.

| Strategy | Noise Init | Training Stages | Spatial Update | Metrics | |
|---|---|---|---|---|---|
| | | | | AbsRel $\downarrow$ | $\delta_1 \uparrow$ |
| Baseline (40k step) | Normal | 1 | - | 0.232 | 0.754 |
| One-Stage Joint | PSNI | 1 | - | 0.221 | 0.763 |
| Two-Stage Unfrozen | PSNI | 2 | Train | 0.215 | 0.769 |
| Two-Stage Frozen | Normal | 2 | Freeze | 0.204 | 0.775 |
| Two-Stage Frozen (Ours) | PSNI | 2 | Freeze | **0.199** | **0.798** |

This suggests that preserving the spatially-adapted weights prevents the model from sacrificing geometric accuracy to satisfy temporal constraints. Crucially, under this optimal protocol, adapting the initial noise via PSNI yields superior results compared to the normal noise baseline. This further confirms that PSNI effectively bridges the geometric mismatch between planar priors and the ERP domain.

**Sensitivity to Inference Resolution.** We further evaluate the inference resolution sensitivity of models in Table 7. Specifically, each fine-tuned model is trained at $640 \times 320$ and evaluated at multiple inference resolutions. We can observe that PVDepth exhibits a similar resolution-dependent trend to DepthCrafter: performance improves substantially from low to medium resolutions and then becomes relatively stable at higher resolutions. Although trained at $640 \times 320$, PVDepth benefits from higher-resolution inference, reducing AbsRel from 0.275 at $640 \times 320$ to 0.199 at $1024 \times 512$ while maintaining comparable $\delta_1$. This suggests that PVDepth is not overly sensitive to inference at resolutions different from the training resolution.

### 5.4. Limitations

While PVDepth achieves superior performance in panoramic video depth estimation, we acknowledge certain limitations regarding inference efficiency. As shown in Table 8, we report inference latency, GPU memory, FLOPs, and accuracy for different methods at $1024 \times 512$ resolution using 110-frame videos. Compared with the baseline DepthCrafter (Hu et al., 2025), PVDepth achieves better depth accuracy with nearly identical GPU memory usage, but introduces higher FLOPs and latency mainly due to the bi-directional projection between ERP and Cube features. Future work could address this by exploring more efficient projection algorithms.

Also, our method is slower than UniK3D (Piccinelli et al., 2025). This is expected as our diffusion process involves iterative sampling, in contrast to the efficient single-pass inference of UniK3D. However, compared to ViPE (Huang et al., 2025), which applies test-time optimization on top of UniK3D to ensure temporal consistency, our PVDepth is faster and more stable. Matrix-3D (Yang et al., 2025b) incurs even higher latency due to its reliance on Poisson blending for panoramic stitching and iterative inter-frame alignment. These comparisons further underscore the need to develop end-to-end video depth models.

In addition, PVDepth still faces limitations under domain shifts and reflective-surface scenarios. Since PanoCARLA contains only outdoor scenes, the generalization of PVDepth to indoor settings may still be limited. We provide qualitative indoor results in Appendix I, but do not report quantitative indoor evaluation due to the lack of publicly available annotated indoor panoramic video datasets. PVDepth also struggles with strong specular reflections, such as reflective water surfaces, as illustrated with qualitative examples in Appendix I. Future work could collect more diverse panoramic RGB-D video data to improve robustness across broader scenarios.

*Table 7.* **Sensitivity to inference resolution.** Each fine-tuned model is trained at $640 \times 320$ and evaluated at different inference resolutions. Each entry reports AbsRel $\downarrow$, with $\delta_1 \uparrow$ shown in parentheses. Both metrics are averaged over the three evaluation settings. ZS and FT denote Zero-Shot and Fine-Tuned models.

| Resolution | DepthCrafter$_{ZS}$ | DepthCrafter$_{FT}$ | PVDepth |
|---|---|---|---|
| $384 \times 192$ | 0.679 (0.473) | 0.493 (0.709) | 0.455 (0.738) |
| $640 \times 320$ | 0.351 (0.571) | 0.354 (0.761) | 0.275 (0.801) |
| $768 \times 384$ | 0.310 (0.570) | 0.263 (0.769) | 0.212 (0.806) |
| $896 \times 448$ | 0.292 (0.577) | 0.254 (0.763) | 0.207 (0.802) |
| $1024 \times 512$ | 0.290 (0.567) | 0.232 (0.754) | 0.199 (0.798) |

*Table 8.* **Inference Efficiency.** Latency is the average inference time per frame at $1024 \times 512$ resolution. GPU memory and FLOPs are measured for 110-frame clips at the same resolution. AbsRel and $\delta_1$ are averaged over the three evaluation settings. ViPE shows a range because the convergence behavior of SLAM optimization varies across sequences.

| Metric | UniK3D | ViPE | Matrix-3D | DepthCrafter | PVDepth |
|---|---|---|---|---|---|
| Latency (s/frame) $\downarrow$ | 0.05 | 0.5–3 | 20 | 0.355 | 0.482 |
| GPU Memory (G/clip) $\downarrow$ | 3.84 | 16.8–34.4 | 9.22 | 30.6 | 30.7 |
| FLOPs (PFLOPs/clip) $\downarrow$ | 0.317 | 0.758–2.258 | 2.518 | 1.986 | 2.553 |
| AbsRel $\downarrow$ | 0.592 | 0.670 | 0.494 | 0.232 | 0.199 |
| $\delta_1 \uparrow$ | 0.638 | 0.721 | 0.513 | 0.754 | 0.798 |

## 6. Conclusion

We propose PVDepth, a generative framework that adapts perspective video depth models for panoramic depth estimation. By introducing the large-scale PanoCARLA dataset and designing geometry-aware adaptation mechanisms (Progressive Sphere-aware Noise Initialization and Cube-rectified Temporal Modeling), we effectively address the scarcity of training data and ERP-induced spatiotemporal challenges. Extensive experiments demonstrate PVDepth's superior performance and promising zero-shot generalization. We believe this work not only establishes a strong baseline but also unlocks new possibilities for 3D scene understanding in virtual reality and generative world models.

## Acknowledgements

The study was funded by the Shenzhen Science and Technology Program (KQTD20240729102051063), the National Natural Science Foundation of China under contracts No. 62422602, No. 62372010, No. 62425101, No. 62332002, U25B6003, and the major key project of the Peng Cheng Laboratory (PCL2021A13 and PCL2025A02). Computing support was provided by Pengcheng Cloudbrain. We thank the anonymous reviewers for their constructive feedback. We are also grateful to Xiaodong Wang and Haoran Xu for their helpful early-stage discussions on training duration and data recording settings, respectively.

## Impact Statement

This paper presents work whose goal is to advance the field of Machine Learning. There are many potential societal consequences of our work, none which we feel must be specifically highlighted here.

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

# A. PanoCARLA Dataset

A primary bottleneck hindering research in panoramic video depth estimation is the scarcity of public RGB-D video datasets. To bridge this gap, we devise a comprehensive strategy to curate a large-scale dataset within the CARLA simulator (Dosovitskiy et al., 2017), encompassing diverse environments (Town01–Town07, Town10). Existing approaches, such as Genex (Lu et al., 2024), typically synthesize capture trajectories by interpolating between proximal viewpoints using predefined movement patterns. Consequently, the resulting motions tend to be rigid and short-lived, occasionally even exhibiting physical artifacts like wall-clipping. This fails to meet the demands of advanced algorithms for motion diversity and realism. To capture more natural and smooth trajectories, we adopt a **straightforward yet effective strategy**: mounting the camera directly onto an autonomous vehicle. This allows us to harness the sophisticated navigation policies built into the simulation to acquire highly realistic and physically consistent trajectories.

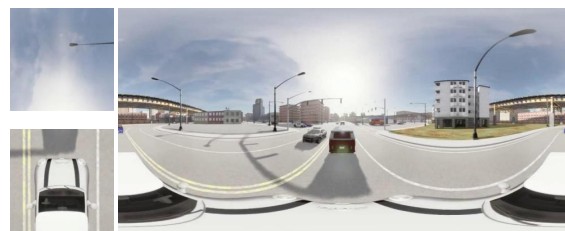
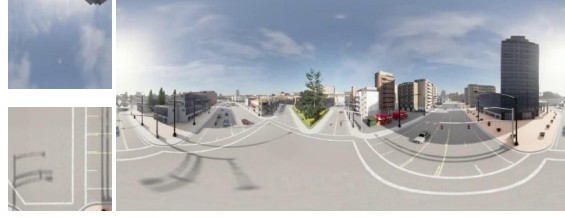

*(a)* With self-occlusion      *(b)* Without self-occlusion

*Figure 5.* Comparison of ERPs with and without vehicle self-occlusion. For each ERP, the left inset shows its up and down cubemap views. In the self-occlusion case (a), the vehicle body blocks a fixed portion of the down view, introducing persistent visual noise.

However, mounting the camera directly onto vehicles introduces a critical issue: *vehicle self-occlusion*. As shown in Fig. 5a, the vehicle body inevitably creates persistent visual "noise." Attempting to remove these artifacts via post-processing (e.g., segmentation and inpainting) is not only computationally expensive but also risks compromising spatiotemporal continuity. To address this, as shown in Fig. 2, we introduce a **Two-Stage Decoupled Data Acquisition Pipeline**. First, we record authentic 6-DoF trajectories from driving vehicles (Stage I) and then re-render the sequences using an invisible "ghost rig" driven by the recorded poses (Stage II). This strategy ensures both physical motion realism and an unobstructed 360° field of view.

## A.1. Stage I: Natural Trajectory Planning

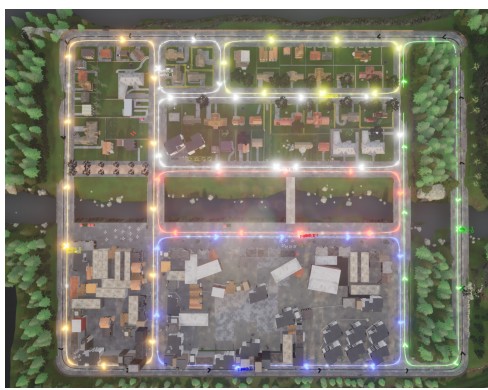
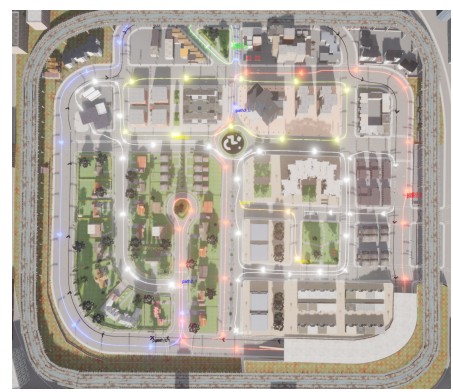

*(a)* Town01 traj.      *(b)* Town03 traj.

*Figure 6.* Examples of recorded trajectory visualizations in Town01 (a) and Town03 (b). The colored loops represent smooth driving paths generated by our pipeline.

The core objective of this phase is to generate smooth and plausible motion trajectories across diverse scenarios. We adopt a hybrid strategy integrating expert planning with autonomous execution:

- **Waypoint Planning:** Experienced human drivers manually designate a series of sparse waypoints on the CARLA map. This strategic placement ensures that the generated routes cover a comprehensive range of road typologies, such as urban arterials, narrow alleys, and multi-level viaducts.

- **Autonomous Execution & Pose Sampling:** These waypoints are fed into CARLA's Navigation Module, which computes a feasible route. The vehicle then navigates the environment using the built-in Autopilot, ensuring the motion adheres to physical constraints and traffic logic. We synchronously sample the vehicle's 6-DoF ground-truth poses at a consistent rate of 20 FPS as it traverses the environment.

- **Human-in-the-Loop Refinement:** All recorded sequences undergo manual visualization and review. Any sequences exhibiting unnatural behavior are flagged for manual intervention, where human experts iteratively adjust the underlying waypoints and re-simulate the drive until the resulting trajectory satisfies our standards for quality and realism.

Through this method, we obtain high-quality motion trajectories that balance global planning and local smoothness. Example trajectory results are visualized in Fig. 6.

### A.2. Stage II: Unobstructed RGB-D Data Recording

This phase aims to render clean scene data using the recorded trajectories while introducing vertical motion complexity. We construct a virtual rigid rig of six synchronized RGB-D cameras. Crucially, this rig is *detached* from any vehicle mesh, driven purely by the pose data from Stage I. This "ghost rig" setup perfectly replicates natural driving motion while eliminating vehicle occlusion.

**Drone-view Roaming.** To simulate more diverse 6-DoF motion, we upgrade the ground-level trajectories to a drone-like roaming perspective. While following the horizontal path, the camera altitude is modulated between $2m$ and $20m$ using smooth randomized functions (e.g., cosine, linear).

**Dynamic Scene & Safety.** We populate the scene with autonomous traffic to introduce dynamic occlusions. To ensure data validity during altitude variation, we equip each camera view with invisible collision sensors. Any frame sequence where the camera clips into environmental structures (e.g., streetlights, buildings) is automatically logged and discarded during post-processing.

### A.3. Panoramic Stitching & Statistics

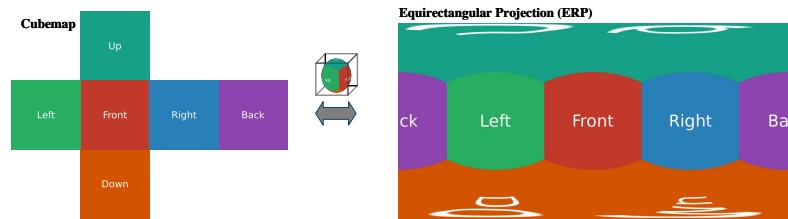

*Figure 7.* Illustration of the conversion from Cubemap to Equirectangular Projection (ERP).

Since CARLA lacks a native panoramic sensor, we employ a six-view Cubemap capture scheme (Front, Back, Left, Right, Up, Down), which is subsequently stitched into the standard Equirectangular Projection (ERP) format (Fig. 7).

**RGB Processing.** We prioritize photorealism for downstream perception tasks. Unlike works that disable auto-exposure to force consistency, we retain these effects to simulate the challenging lighting variations encountered by real-world multi-camera rigs. To mitigate the resulting seams, inspired by real-world panoramic photography (Zhao, 2020), we render each view with a $120°$ FoV to ensure sufficient overlap and apply **feather blending** in overlapping regions.

Formally, let $I_i$ denote the $i$-th perspective RGB image and $M_i$ its corresponding feathering mask, defined in the perspective domain with weights decaying from the image center to the boundaries. Both $I_i$ and $M_i$ are projected to the ERP space via

bilinear sampling, yielding $\tilde{I}_i$ and $\tilde{M}_i$. The final panoramic RGB image $\mathcal{I}$ is obtained by normalized weighted averaging:

$$\mathcal{I} = \frac{\sum_i \tilde{I}_i \odot \tilde{M}_i}{\sum_i \tilde{M}_i}, \tag{4}$$

where $\odot$ denotes element-wise multiplication.

**Depth Processing.** Raw simulator output is planar Z-depth (projected along the optical axis), which is geometrically incompatible with spherical stitching due to divergent optical axes. Direct stitching would cause severe artifacts. Therefore, we implement a pixel-wise calibration step: converting Z-depth to **Euclidean distance** (radial distance from the optical center) before stitching. This ensures that depth values are aligned to a unified spherical metric.

For stitching, although each view is rendered with a $120°$ FoV, we extract only the central $90°$ FoV from each view to form a standard non-overlapping cubemap, since the calibrated depth provides sufficient geometric accuracy. The depth maps are directly reprojected to the ERP format and stitched without blending, resulting in seam-free and geometrically consistent depth.

**Dataset Statistics.** Finally, we employ six RGB cameras and six depth cameras, all configured with a $120°$ FoV and a resolution of $800 \times 800$. This setup enables the generation of high-resolution panoramic frames up to $\mathbf{1200 \times 2400}$. We initially designed 67 comprehensive roaming trajectories across diverse towns. Following the automated collision detection and safety pruning process, these trajectories were segmented into 117 high-quality video clips to ensure that every frame maintains geometric integrity without environment clipping.

In total, PanoCARLA comprises **126.7 k** synchronized RGB-D frames and covers a total roaming distance of **50.61 km**. On average, each clip contains approximately **1,083** frames and traverses **432.58** meters. For each frame, we provide the stitched RGB panorama, panoramic depth map, and camera extrinsics. Additionally, we will release the raw six-view RGB-D data to support future research on improved stitching algorithms. Detailed statistics for each town are presented in Table 9. For more qualitative examples, we provide additional visualizations of the PanoCARLA dataset in Fig. 15, displaying RGB-D pairs across various scenarios.

*Table 9.* Statistics of different towns in the PanoCARLA dataset. T denotes Town index.

| STATISTIC | T01 | T02 | T03 | T04 | T05 | T06 | T07 | T10 | TOTAL |
|---|---|---|---|---|---|---|---|---|---|
| # CLIPS | 18 | 14 | 22 | 13 | 19 | 9 | 14 | 8 | **117** |
| # FRAMES | 15512 | 6626 | 17822 | 23531 | 25436 | 15788 | 14899 | 7082 | **126696** |
| DISTANCE (KM) | 5.82 | 2.53 | 6.46 | 10.49 | 9.63 | 7.45 | 5.71 | 2.52 | **50.61** |

# B. Preliminaries: SVD and DepthCrafter

Our framework is built upon DepthCrafter (Hu et al., 2025), which repurposes the architecture of Stable Video Diffusion (SVD) (Blattmann et al., 2023) for high-quality video depth estimation. In this section, we first review the formulation of the underlying SVD model and then describe the specific adaptations introduced by DepthCrafter.

**Stable Video Diffusion (SVD).** SVD is a latent video diffusion model that operates in a compressed space learned by a Variational Autoencoder (VAE) to reduce computational complexity. Given a data sample $\mathbf{x}$ (e.g., a video clip), the VAE encoder $\mathcal{E}$ maps it to a latent representation $\mathbf{z}_0 = \mathcal{E}(\mathbf{x})$. During training, a forward process progressively corrupts the clean latent $\mathbf{z}_0$ with Gaussian noise $\boldsymbol{\epsilon} \sim \mathcal{N}(\mathbf{0}, \mathbf{I})$. At time step $t$, the noisy latent $\mathbf{z}_t$ is formulated as:

$$\mathbf{z}_t = \mathbf{z}_0 + \sigma_t \boldsymbol{\epsilon}, \tag{5}$$

where $\sigma_t$ controls the noise level. A denoiser network $D_\theta$ is trained to recover the clean $\mathbf{z}_0$ from $\mathbf{z}_t$, conditioned on context $\mathbf{c}$. The training objective minimizes the weighted squared error:

$$\mathcal{L} = \mathbb{E}_{\mathbf{z}_0, \mathbf{z}_t, t, \mathbf{c}} \left[ w(\sigma_t) \| D_\theta(\mathbf{z}_t; \sigma_t, \mathbf{c}) - \mathbf{z}_0 \|_2^2 \right], \tag{6}$$

where $w(\sigma_t)$ is a noise-dependent weighting function.

**Adaptation in DepthCrafter.** Since SVD is originally designed for image-to-video generation, DepthCrafter introduces two key modifications to adapt it for video-to-depth estimation:

- **Latent Space Adaptation.** The pre-trained VAE in SVD expects 3-channel RGB inputs. To process single-channel depth maps $\mathbf{d}$, DepthCrafter replicates the depth map three times before encoding: $\mathbf{z}^{(d)} = \mathcal{E}(\mathbf{d})$. During decoding, the three output channels from the VAE decoder are averaged to reconstruct the final single-channel depth sequence.

- **Video Conditioning.** To ensure the generated depth is strictly aligned with the input video, DepthCrafter modifies the conditioning mechanism. Instead of conditioning on a single image (as in the original SVD), it conditions on the entire input video sequence $\mathbf{v}$. Specifically, the video latent $\mathbf{z}^{(v)} = \mathcal{E}(\mathbf{v})$ is concatenated frame-wise with the noisy depth latent $\mathbf{z}_t$ along the channel dimension. Additionally, CLIP embeddings extracted from each video frame are injected via cross-attention to provide semantic guidance.

## C. Details of Progressive Sphere-aware Noise Initialization

We provide the detailed procedure of our Progressive Sphere-aware Noise Initialization in Algorithm 1. The decay duration $T_{end}$ is set to 5K steps, and the residual weight $w_{final}$ is set to 0.1 to retain a minimal level of randomness in the planar domain.

---

**Algorithm 1** Progressive Sphere-Aware Noise Initialization

---

1: **Input:** Global training step $t$, Annealing duration $T_{end}$, Final normal weight $w_{final}$, ERP latent size $H_l \times W_l$
2: **Output:** Initial noise $\epsilon$
3: *// 1. Calculate Linear Decay Progress*
4: $\rho \leftarrow \min(1.0, \ t/T_{end})$
5: *// 2. Determine Weights*
6: $w_n \leftarrow 1.0 - \rho \cdot (1.0 - w_{final})$
7: $w_s \leftarrow \sqrt{1.0 - w_n^2}$
8: *// 3. Sample Independent Noise Components*
9: $D \leftarrow W_l/4$
10: $\epsilon_{sphere} \sim \mathcal{N}(\mathbf{0}, \mathbf{I}) \in \mathbb{R}^{D \times D \times D}$
11: $\epsilon_s \leftarrow \mathcal{P}_{S \to E}(\epsilon_{sphere}; H_l, W_l)$   *// Project 3D noise to ERP by trilinear sampling*
12: $\epsilon_n \sim \mathcal{N}(\mathbf{0}, \mathbf{I}) \in \mathbb{R}^{H_l \times W_l}$   *// Sample noise directly in ERP*
13: *// 4. Fuse Components*
14: $\epsilon \leftarrow w_s \cdot \epsilon_s + w_n \cdot \epsilon_n$
15: **Return** $\epsilon$

---

**Sphere-aware Noise Construction.** In Algorithm 1, the sphere-aware noise is constructed through a fixed geometry-based projection. For an ERP latent of size $H_l \times W_l$, we first sample a standard Gaussian noise volume $\epsilon_{sphere} \in \mathbb{R}^{D \times D \times D}$ in a three-dimensional grid, where we set $D = W_l/4$ following the common ERP-to-cubemap resolution correspondence.

The projection operator $\mathcal{P}_{S \to E}$ maps this 3D noise volume onto the ERP latent grid. Specifically, for each ERP latent pixel, we compute its latitude $\theta \in [-\pi/2, \pi/2]$ and longitude $\phi \in [-\pi, \pi]$, and convert them into a unit 3D direction:

$$(x, y, z) = (\cos\theta \cos\phi, \ \cos\theta \sin\phi, \ \sin\theta). \tag{7}$$

The resulting direction specifies the corresponding sampling location in the 3D noise volume. Since the location generally does not lie exactly on discrete grid points, we obtain the ERP noise value by **trilinear interpolation**. This fixed projection introduces no learnable parameters.

For a fixed latent resolution, the ERP-to-3D coordinate map is deterministic, so it can be precomputed once and cached for both training and inference.

Through the above process, the generated ERP noise can be interpreted as the equirectangular representation of a Gaussian noise signal defined with respect to the underlying spherical geometry, while introducing negligible additional overhead in practice.

# D. Implementation Details

**Training details of PVDepth.** Our PVDepth is initialized with pre-trained DepthCrafter (Hu et al., 2025) weights. To ensure stable convergence, we employ a two-stage training strategy: Stage I (Spatial Adaptation): We fine-tune the spatial layers for 20k steps using our Progressive Sphere-aware Noise Initialization (PSNI). This stage focuses on adapting the U-Net spatial priors to the ERP's spherical geometry. Stage II (Temporal Rectification): We introduce the Cube-rectified Temporal Modeling module and fine-tune *only* the newly added projection layers for another 20k steps. Crucially, during this stage, the noise initialization is fixed to the final state of PSNI (i.e., using the constant residual planar weight $w_{final}$ without further annealing).

In all stages, we use the AdamW optimizer with a learning rate of $5 \times 10^{-6}$ and a batch size of 8. Leveraging the robust temporal priors established by DepthCrafter's extensive training on long-sequence data, we freeze the original temporal layers throughout our training. This strategy allows us to preserve the backbone's inherent capacity to generalize across variable video lengths. Consequently, training with a fixed clip length of 6 frames proves effective in adapting the model to the panoramic domain. To ensure adaptability to diverse motion dynamics, we randomly sample frame rates from $[1, 20]$ at each training step. Consistent with the training protocol of DepthCrafter, we perform training at a resolution of $640 \times 320$ for computational efficiency, while evaluation uses the common panoramic resolution of $1024 \times 512$ (Wang & Liu, 2024; Cao et al., 2025; Li et al., 2025a). All experiments are conducted on NVIDIA A100 GPUs.

**Baseline Settings.** To ensure a fair comparison, the Baseline model follows a strictly aligned training protocol. Specifically, we fine-tune the original DepthCrafter model on the PanoCARLA dataset for 20k steps. We empirically verified that extending training to 40k steps yields negligible gains (see Table 3). Consistent with our PVDepth, we fine-tune the spatial layers while keeping the temporal layers frozen, but employ *standard Gaussian noise initialization* instead of our sphere-aware strategy.

**Inference Settings.** During inference, consistent with DepthCrafter, we use a 5-step denoising schedule. To enable arbitrary-length video depth estimation, we adopt the sliding window inference strategy, as detailed in (Hu et al., 2025).

# E. Evaluation Metrics

Following evaluation protocols in previous video depth estimation works (Hu et al., 2025; Chen et al., 2025; Yang et al., 2025a), we employ metrics including Absolute Relative Error (AbsRel) and Threshold Accuracy ($\delta_1$). Let $d_i$ and $\hat{d}_i$ denote the ground truth and predicted depth values at valid pixel $i$, and $N$ be the total number of valid pixels. The AbsRel metric is formulated as:

$$\text{AbsRel} = \frac{1}{N} \sum_{i=1}^{N} \frac{|d_i - \hat{d}_i|}{d_i}. \tag{8}$$

The threshold accuracy $\delta_1$ is defined as:

$$\delta_1 = \frac{1}{N} \sum_{i=1}^{N} \mathbb{I}\left( \max\left( \frac{d_i}{\hat{d}_i}, \frac{\hat{d}_i}{d_i} \right) < 1.25 \right). \tag{9}$$

where $\mathbb{I}(\cdot)$ is the indicator function. Consistent with normal outdoor settings (Hu et al., 2025), we cap the maximum evaluation depth at 80m.

In addition, following (Hu et al., 2025; Chen et al., 2025; Yang et al., 2025a), we perform **affine-invariant alignment** to eliminate the scale and shift ambiguity inherent in monocular depth estimation. Specifically, we align the predicted depth to the ground truth by optimizing a scale $s$ and shift $t$. Crucially, to faithfully reflect temporal consistency, we perform this alignment globally, optimizing a single pair of $(s, t)$ for the *entire video clip*. This stands in contrast to per-frame alignment, where independent adjustments for each frame can artificially conceal temporal flickering artifacts.

# F. Quantitative Reprojection-based Geometric Consistency Evaluation

Due to the lack of publicly available real-world panoramic video datasets with depth annotations, we cannot directly evaluate the accuracy of predicted depth in real-world scenarios. Therefore, this section introduces a reprojection-based geometric consistency evaluation protocol for quantitatively assessing existing methods under real-world conditions. All experiments

are conducted on **GenEx Real-World**, a real panoramic video subset of **GenEx-DB** (Lu et al., 2024). This dataset is well suited for reprojection-based evaluation, as its scenes are predominantly static campus environments, satisfying a key assumption required for geometric consistency analysis.

### F.1. Experimental Setup

We randomly select 50 video clips from the GenEx Real-World dataset for evaluation. Since GenEx Real-World does not provide ground-truth camera poses for each frame, we estimate the camera trajectory for each clip using the SLAM module provided in ViPE (Huang et al., 2025) (see Appendix G). The estimated poses are treated as reference poses and are used solely for geometric reprojection during evaluation.

Although the estimated camera poses may not be perfectly accurate, ViPE jointly optimizes camera poses and depth estimates, resulting in a self-consistent depth–pose configuration. We therefore align the predicted depths of all methods to the depth output of ViPE to resolve scale ambiguity across different depth prediction methods and ensure a fair comparison. Specifically, for each video clip, a single **global scale and shift** is estimated over the entire sequence and applied uniformly to all frames, following the same protocol described in Appendix E. This video-level alignment implicitly requires depth predictions to remain accurate and temporally consistent across frames.

For each video clip, we select multiple anchor frames at fixed temporal intervals, and for each anchor frame, only the predicted depth of that frame is used to back-project the panoramic image into a 3D point cloud. Using the estimated camera poses from ViPE, the point cloud is then forward-projected to subsequent panoramic frames. For computational efficiency, both the anchor frames and temporal offsets are subsampled, with anchor frames selected at frame indices $\{0, 20, 40, 60, 80\}$, and reprojection evaluated at temporal offsets $\{5, 10, 20, 30, 40, 50, 60, 70, 80, 90\}$.

Although GenEx Real-World is captured in near-static environments, the camera holder remains visible in the panoramic view and constitutes a dynamic object, which violates the static-scene assumption required for reprojection-based evaluation. To alleviate its influence, we mask the bottom 15% region of panoramic images, excluding the corresponding pixels in the source frame from point cloud generation and masking the same region in the target frame during metric computation. Reprojection-based geometric consistency is then evaluated by comparing the synthesized target views with the corresponding real panoramic frames over valid regions only.

### F.2. Evaluation Metric

Let $I_t \in \mathbb{R}^{H \times W \times 3}$ denote the real target panoramic image at time $t$, and let $\hat{I}_t$ denote the synthesized image obtained via reprojection. Let $M_t \in \{0, 1\}^{H \times W}$ be a binary mask indicating valid pixels for evaluation, and define the set of valid pixels as $\Omega = \{i \mid M_t(i) = 1\}$.

We use the Structural Similarity Index (SSIM) as the evaluation metric to measure reprojection-based geometric consistency. SSIM evaluates perceptual similarity between two images by comparing luminance, contrast, and structural information. Given two local image patches $x$ and $y$ extracted from $I_t$ and $\hat{I}_t$, SSIM is defined as

$$\text{SSIM}(x, y) = \frac{(2\mu_x \mu_y + C_1)(2\sigma_{xy} + C_2)}{(\mu_x^2 + \mu_y^2 + C_1)(\sigma_x^2 + \sigma_y^2 + C_2)}, \tag{10}$$

where $\mu_x$ and $\mu_y$ denote the mean intensities of $x$ and $y$, $\sigma_x^2$ and $\sigma_y^2$ are the corresponding variances, and $\sigma_{xy}$ denotes their covariance. The constants $C_1$ and $C_2$ are used to stabilize the division. The final SSIM score is obtained by averaging SSIM values over all valid pixels in $\Omega$.

The final reported scores are obtained by averaging SSIM over all evaluated frames across all video clips.

### F.3. Results and Analysis

Table 10 reports the overall reprojection-based SSIM results on the GenEx Real-World dataset. Notably, PVDepth achieves slightly higher SSIM than ViPE, even though ViPE internally estimates and optimizes camera poses used in the evaluation, indicating improved geometric consistency under the proposed evaluation protocol.

The baseline DepthCrafter also benefits from temporal modeling and yields more consistent depth than single-frame methods; however, its performance remains limited due to the domain gap between perspective training data and panoramic imagery. In contrast, the single-frame depth estimation method DA-2 performs relatively poorly, which can be attributed to significant

scale variation across frames, as a single scale and shift are applied at the video level during evaluation.

*Table 10.* Reprojection-based geometric consistency evaluation on the GenEx Real-World dataset using SSIM. Higher values indicate better geometric consistency.

| Metric | ViPE (Huang et al., 2025) | DepthCrafter (Hu et al., 2025) | DA-2 (Li et al., 2025a) | PVDepth |
|---|---|---|---|---|
| SSIM ↑ | 0.4107 | 0.3290 | 0.3147 | **0.4194** |

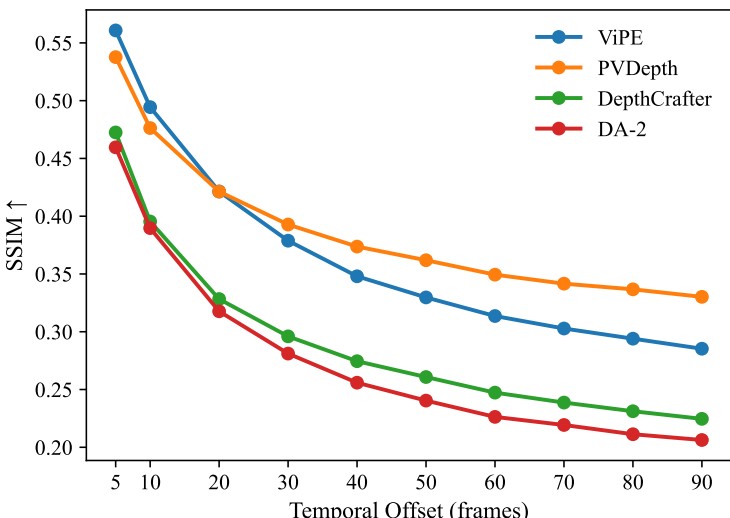

*Figure 8.* **Reprojection-based geometric consistency under varying temporal offsets.** We report SSIM scores of different methods under varying temporal offsets between the source and target frames. While ViPE achieves strong performance at small temporal offsets due to test-time optimization, its performance degrades more rapidly as the temporal offset increases. In contrast, PVDepth exhibits more stable performance across larger temporal offsets, indicating stronger long-range temporal consistency.

We further analyze the performance of different methods under varying temporal offsets between the source and target frames. As shown in Fig. 8, for small temporal offsets, ViPE achieves the highest SSIM, benefiting from its test-time optimization that aligns depth predictions across nearby frames in conjunction with the estimated camera poses. However, as the temporal offset increases, the performance of ViPE degrades more rapidly than that of PVDepth. In contrast, PVDepth exhibits more stable performance at larger temporal offsets, indicating stronger long-range temporal consistency, which contributes to its higher overall SSIM performance. This observation suggests that relying solely on test-time optimization for cross-frame depth alignment is insufficient for distant frames, further highlighting the importance of training end-to-end panoramic video depth estimation models.

To provide more intuitive insights beyond aggregate SSIM scores, we further visualize reprojection results at a representative temporal offset. Fig. 9 shows qualitative comparisons at a temporal offset of 40 frames, where synthesized target views are rendered by forward-projecting point clouds from an anchor frame. Gray regions correspond to masked invalid areas and are excluded from evaluation. It can be observed that DA-2 exhibits the largest discrepancy from the target frame, likely due to insufficient depth–pose alignment. Specifically, reprojection patterns originating from the bottom region of the anchor frame appear as two gray semicircular regions on both sides in the synthesized views, which are clearly missing in the DA-2 result. DepthCrafter achieves better alignment but introduces severe geometric distortions, which can be attributed to the domain gap between perspective-based training data and panoramic imagery. In contrast, PVDepth produces reprojections that are more geometrically consistent with the target frame. These results highlight the potential of applying PVDepth to downstream tasks that rely on geometrically consistent reprojection, such as 3D reconstruction or point-cloud–guided panoramic world generation.

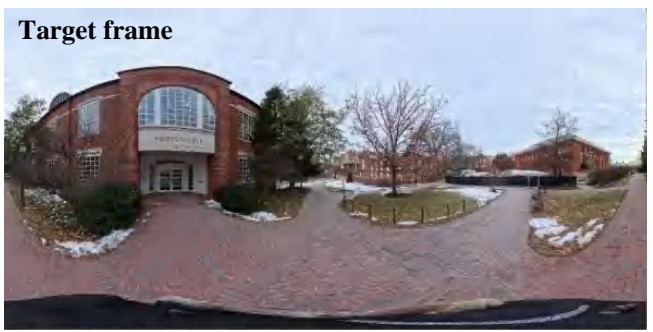

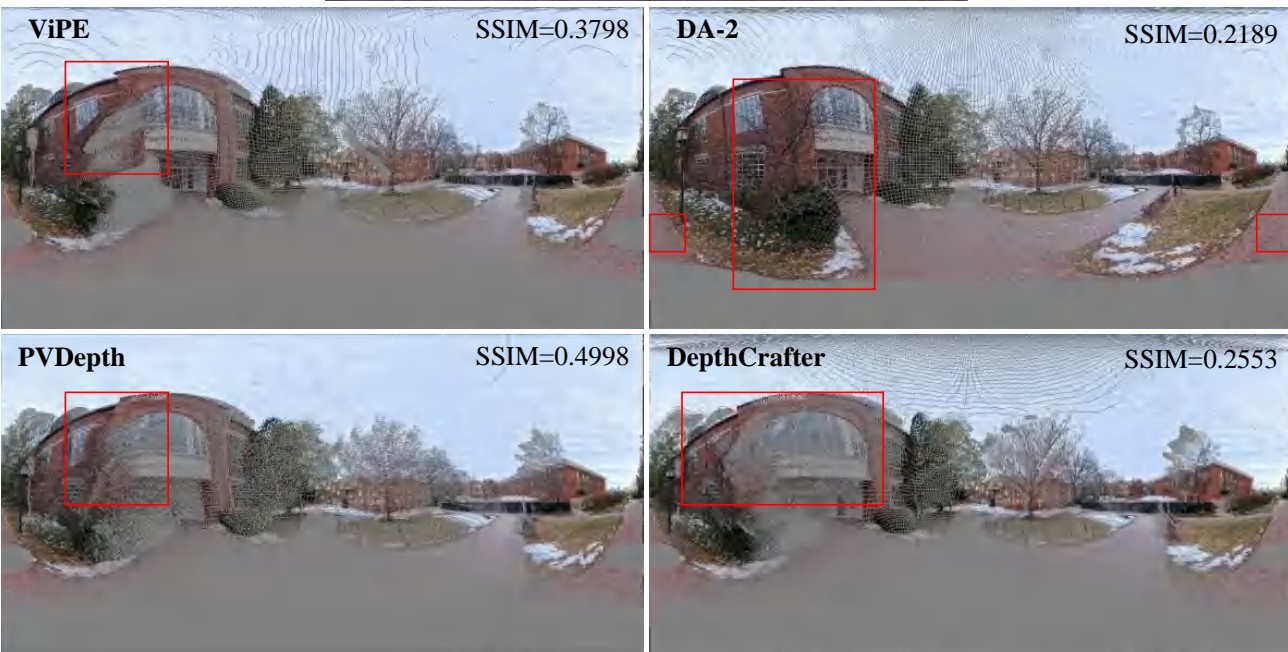

*Figure 9.* **Qualitative visualization of reprojection-based geometric consistency.** The top row shows the target panoramic frame. The second and third rows show synthesized target views obtained by forward-projecting point clouds from an anchor frame at a temporal offset of 40 frames using different methods. Gray regions indicate invalid areas that are masked out during evaluation. Compared to other methods, DA-2 exhibits the largest discrepancy from the target frame due to poor depth–pose alignment, while DepthCrafter suffers from noticeable geometric distortions caused by the domain gap between perspective training data and panoramic imagery. In contrast, PVDepth produces more geometrically consistent reprojections that better align with the target frame.

## F.4. Additional Validation with an Alternative Pose Source

To further examine the robustness of our real-world evaluation and address the concern of potential circular dependency caused by using ViPE-derived poses and scale alignment, we conduct an additional evaluation with an alternative pose source. Specifically, we use MapAnything (Keetha et al., 2025) to estimate camera poses and depth for the same GenEx Real-World clips. Since MapAnything does not directly support ERP input, we first project each panoramic video into a front-view cubemap sequence, estimate depth and camera poses with MapAnything, and then project the estimated depth back to the ERP domain together with the corresponding valid-region mask. The resulting depth and poses are used for scale alignment and reprojection-based evaluation, following the same SSIM metric as in the main real-world evaluation.

As shown in Table 11, PVDepth achieves the highest SSIM under both ViPE-based and MapAnything-based pose sources. This suggests that the observed real-world improvement is not solely caused by the ViPE-based evaluation protocol, further supporting the robustness of our real-world comparison.

*Table 11.* Reprojection-based geometric consistency on the GenEx Real-World dataset under different pose sources, measured by SSIM. Higher values indicate better geometric consistency.

| Pose Source | ViPE (Huang et al., 2025) | DepthCrafter (Hu et al., 2025) | DA-2 (Li et al., 2025a) | PVDepth |
|---|---|---|---|---|
| MapAnything-based | 0.3460 | 0.3023 | 0.2572 | **0.3767** |
| ViPE-based | 0.4107 | 0.3290 | 0.3147 | **0.4194** |

## G. Detailed Configurations for Comparison methods

Given the absence of existing end-to-end panoramic video depth estimation models, we compare PVDepth against three categories of approaches that can be used for panoramic video depth estimation: image-based panoramic depth models, post-processing alignment frameworks, and perspective video depth models.

**Image Depth-based Methods.** We included representative methods: DA-2 (Li et al., 2025a), UniK3D (Piccinelli et al., 2025), PanoDA (Cao et al., 2025), and DepthAnywhere (Wang & Liu, 2024). Since these algorithms are inherently designed for images, they cannot directly process video sequences. Therefore, we perform inference frame-by-frame and concatenate the outputs to form the video depth. To further ensure a fair comparison, we have also fine-tuned DepthAnywhere (Wang & Liu, 2024) using its official training codebase on our dataset.

**Post-processing Alignment Methods.** These methods (Yang et al., 2025b; Huang et al., 2025) aim to enforce temporal consistency on top of single-frame predictions through optimization or geometric constraints.

- **Matrix-3D** (Yang et al., 2025b): This method first generates single-frame panoramic depth maps by stitching predictions from a perspective depth model via Poisson blending. Subsequently, it enforces temporal alignment through inter-frame reprojection and global least-squares optimization, which explicitly requires known camera extrinsics as input. Note that in our evaluation, we directly provide Matrix-3D with the ground-truth camera extrinsics available in the dataset. However, this alignment mechanism strictly relies on a static scene assumption, and may break down in the presence of dynamic objects or non-rigid scene changes.

- **ViPE** (Huang et al., 2025): ViPE adopts a "predict-then-optimize" paradigm rooted in a dense SLAM framework. It integrates per-frame depth predictions from UniK3D (Piccinelli et al., 2025) as regularization terms into a dense Bundle Adjustment (BA) to jointly optimize camera poses and scene geometry. To mitigate interference from moving objects, ViPE employs semantic masks from GroundingDINO and SAM to explicitly exclude dynamic regions during optimization.

Note that **HoloTime** (Zhou et al., 2025) also attempts panoramic video depth estimation in pipeline, but they fundamentally rely on a *static viewpoint assumption* (i.e., fixing the background geometry to match the first frame). This constraint renders them incapable of handling the significant 6-DoF camera motion present in PanoCARLA. Consequently, they were excluded from our comparison.

**Perspective Video Depth Estimation.** This category of methods is capable of processing video inputs in an end-to-end manner. We evaluated them under three settings:

1. **Zero-Shot:** We evaluated Video Depth Anything (Chen et al., 2025), Depth Any Video (Yang et al., 2025a), and DepthCrafter (Hu et al., 2025) by directly performing inference on the ERP sequences using their officially released pre-trained weights.

2. **Fine-Tuned Baseline:** To establish a strong baseline, we selected **DepthCrafter** (Hu et al., 2025) and fine-tuned it on our PanoCARLA dataset. For a fair comparison, this fine-tuning was conducted using the exact same training schedule and iteration count as our proposed PVDepth.

3. **Cube-based Strategy:** In this strategy (referred to as DepthCrafter (Cube-based) in Tab. 2), the panoramic video is decomposed into six cubemap sequences, which are processed independently and then re-projected back to the

ERP format. We evaluate both the zero-shot performance of DepthCrafter under this cubemap formulation and a fine-tuned variant trained on the PanoCARLA dataset. During fine-tuning, instead of directly using ERP videos, each panoramic video is decomposed into six cubemap videos, which are jointly used as inputs to ensure effective training and cross-view consistency. Visualizations of the zero-shot and fine-tuned results are shown in Fig. 10.

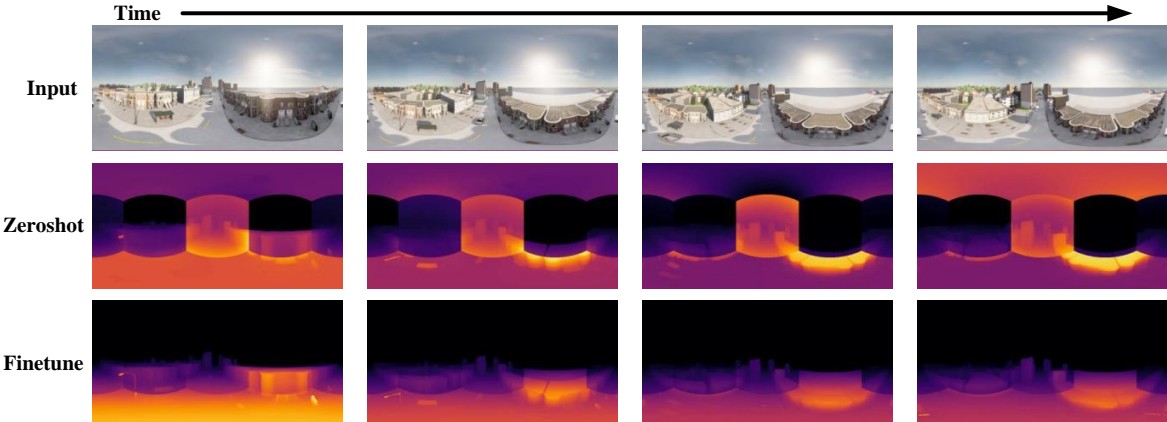

*Figure 10.* Qualitative visualization of the cube-based strategy after re-projecting cubemap predictions back to the ERP format. The first row shows the input RGB panoramic frames. The second and third rows present the corresponding depth predictions obtained by DepthCrafter in the zero-shot and fine-tuned settings, respectively. Although depth predictions are internally consistent within each cubemap region, significant scale discrepancies can be observed across different cubemap regions after re-projection, leading to visible seams in the ERP panorama. Notably, such cross-region scale inconsistencies persist even after fine-tuning.

## H. Additional Frame-wise Qualitative Results

Fig. 11 shows frame-wise qualitative comparisons on in-the-wild panoramic videos, where frames are sampled at a fixed temporal interval of 25 frames (i.e., Frame 0, 25, 50, and 100) from the same panoramic sequence.

It can be observed that in the input RGB panoramas, the visual content in the **lower-right region** remains largely unchanged across these frames. However, depth predictions from *single-frame methods* such as DA-2 (Li et al., 2025a) and UniK3D (Piccinelli et al., 2025) exhibit noticeable fluctuations in this region, indicating temporal instability despite minimal appearance variation. ViPE (Huang et al., 2025) alleviates part of this jitter through iterative alignment (e.g., around Frame 75), yet residual depth oscillations are still observable over time.

In contrast, PVDepth produces more consistent depth predictions across frames, demonstrating superior robustness to long-term temporal evolution in open-world panoramic videos. Fig. 12 and Fig. 14 present more frame-wise qualitative results of PVDepth on in-the-wild panoramic videos.

## I. Additional Limitations

As shown in Fig. 12, PVDepth is able to produce plausible depth predictions on real-world indoor panoramic videos, despite being trained exclusively on outdoor data from PanoCARLA. We conjecture that this generalization is partly attributable to the pretrained priors from DepthCrafter (Hu et al., 2025), and even SVD (Blattmann et al., 2023), which have seen indoor scenes during pretraining. However, due to the lack of publicly available RGB-D indoor panoramic video datasets, this observation cannot be quantitatively evaluated at present. A potential direction for future work is to incorporate indoor panoramic data to further improve the model's robustness across diverse scene types.

In addition, as illustrated in Fig. 13, while PVDepth produces plausible depth estimates on calm water regions (top row), it struggles in the case with strong reflections (bottom row), where reflected structures on the water surface are incorrectly interpreted as physical geometry. This limitation is likely caused by the absence of such patterns in the training data. A possible remedy is to incorporate reflection-aware training strategies, such as fine-tuning on datasets containing reflective

surfaces or augmenting the training data with synthetic reflection effects.

Interestingly, we observe that multiple existing depth models (Li et al., 2025a; Piccinelli et al., 2025; Huang et al., 2025; Hu et al., 2025) also fail in this case, producing either invalid outputs or implausible depth predictions. This suggests that handling scenes with strong specular reflections appears to be a challenging scenario for current depth estimation models and warrants further investigation in future work.

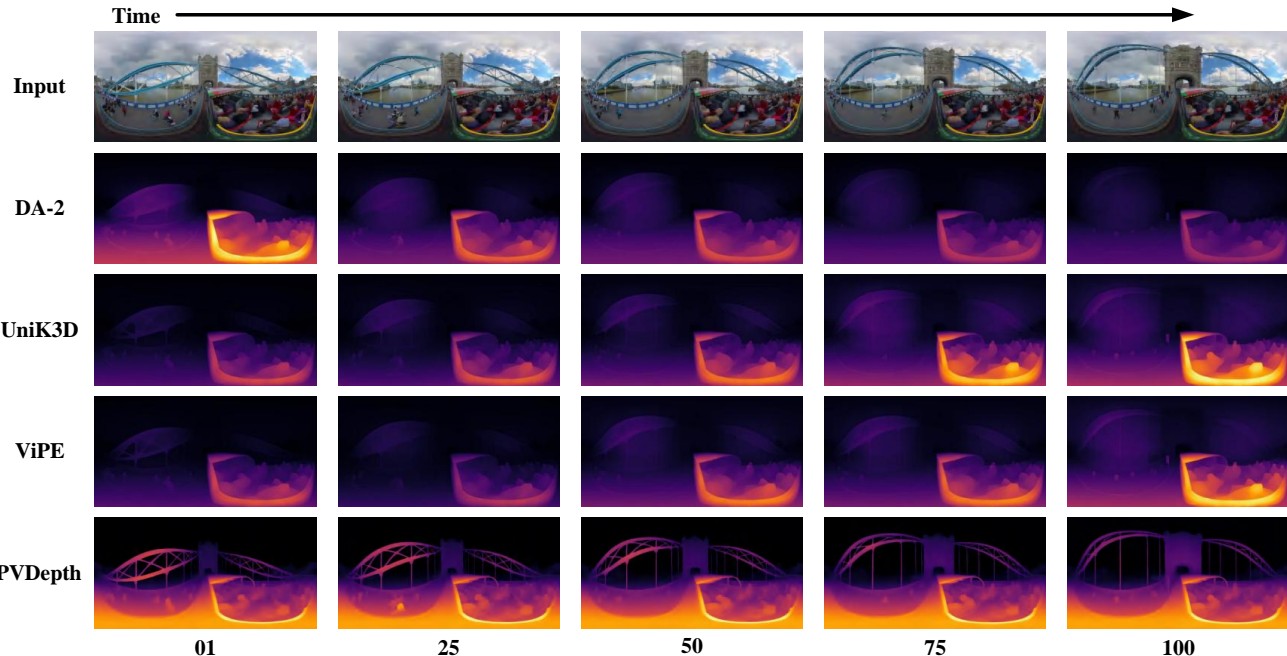

*Figure 11.* Frame-wise qualitative comparisons on in-the-wild panoramic video depth estimation. Columns show individual frames sampled at different temporal positions (Frame 0, 25, 50, 75, and 100) from the same video sequence, while rows correspond to the input RGB panorama and depth predictions from different methods.

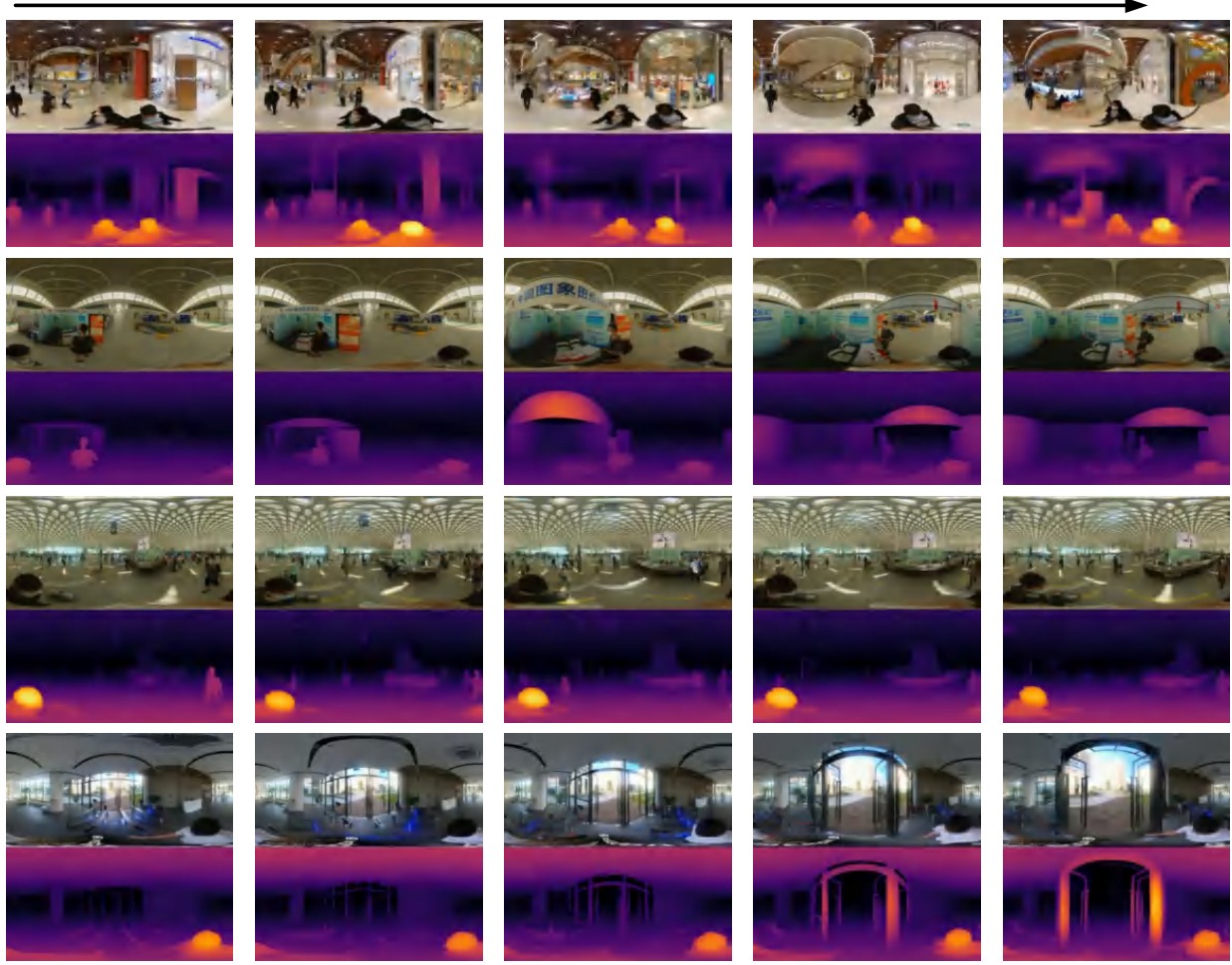

*Figure 12.* Qualitative visualizations of PVDepth on real-world **indoor** panoramic videos.

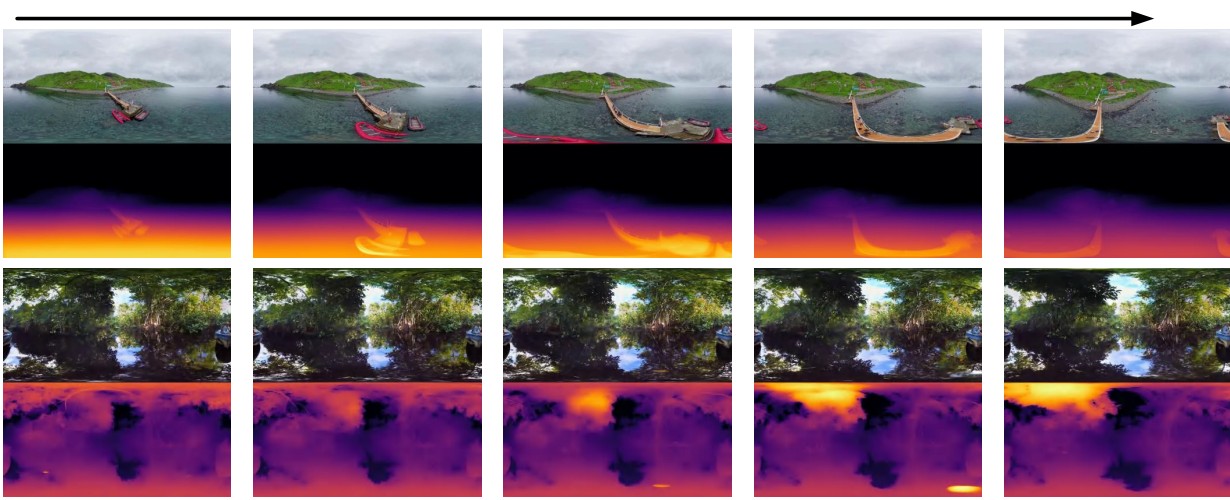

*Figure 13.* **Failure cases of PVDepth on reflective water surfaces**. While PVDepth produces plausible depth estimates on calm water regions (top row), it struggles in scenarios with strong reflections (bottom row), where reflected structures on the water surface are incorrectly interpreted as physical geometry.

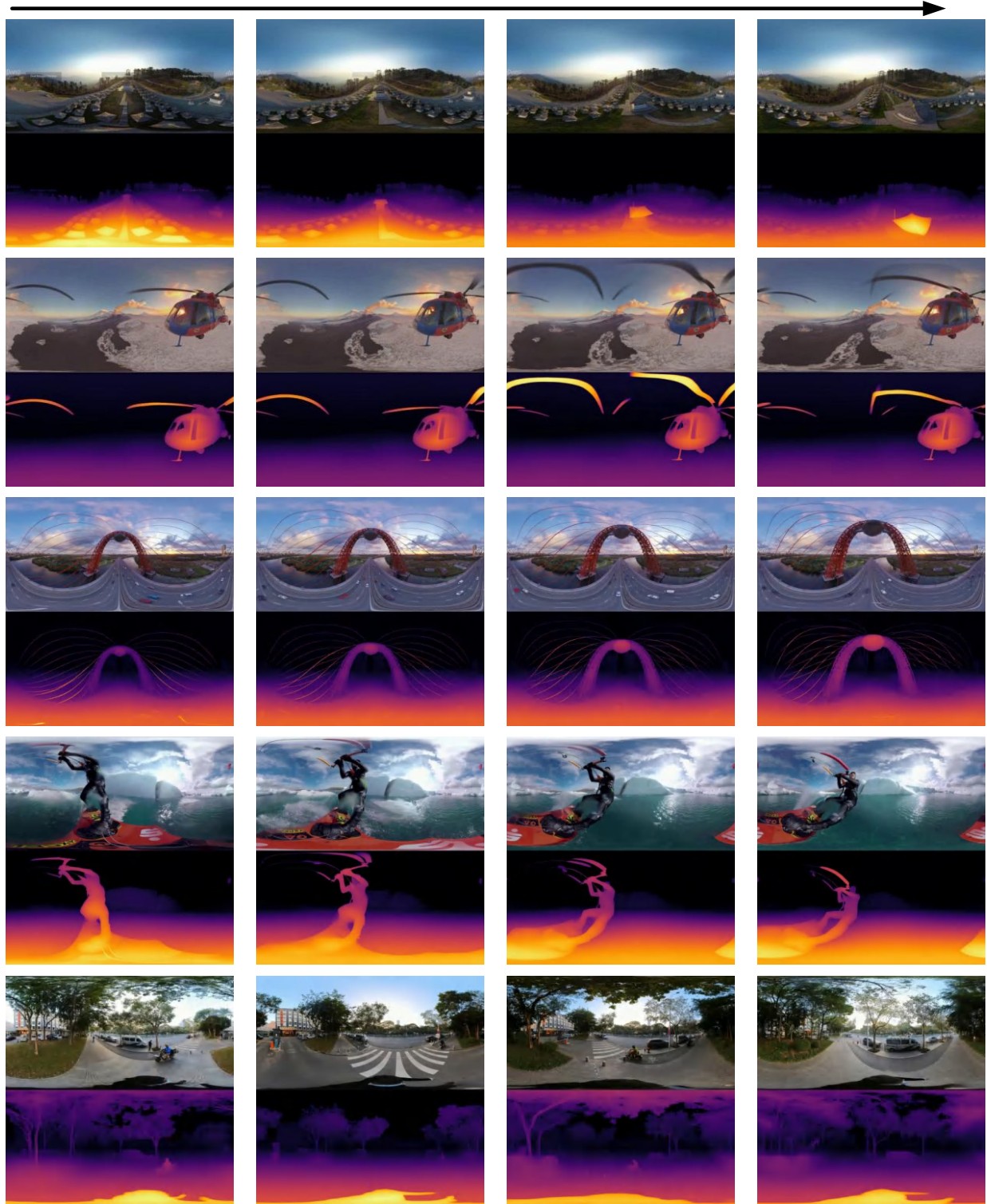

*Figure 14.* Additional frame-wise qualitative visualization results of **PVDepth** on in-the-wild panoramic videos.

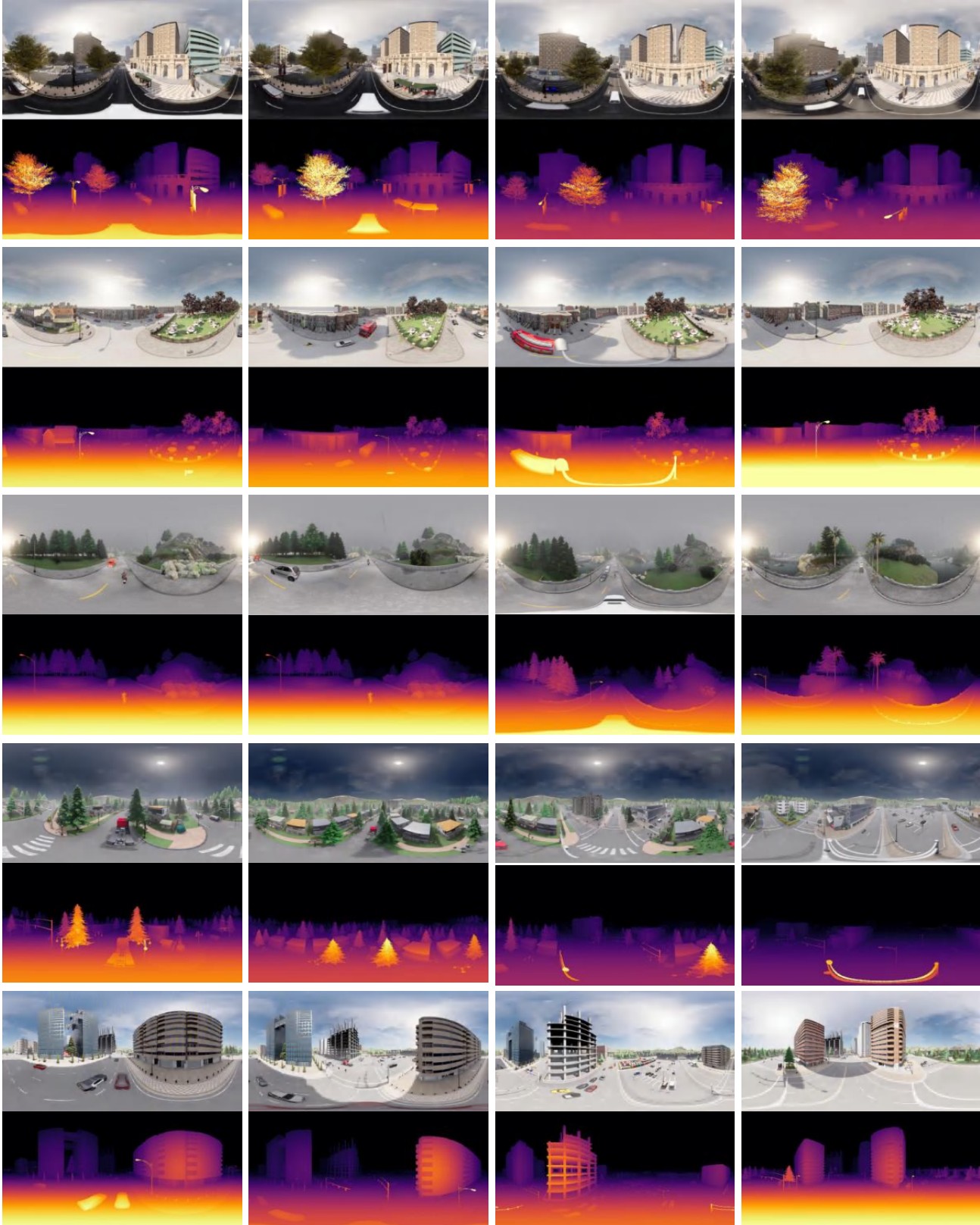

*Figure 15.* **Additional visualizations of the PanoCARLA dataset.** We present sample RGB-D pairs from the collected sequences. The top rows display the stitched RGB panoramas across different environments, while the bottom rows show the corresponding depth visualizations. *For better visualization of fine-grained details, each frame is visualized individually rather than using a sequence-level visualization.*

