# OpenReview forum: "PVDepth: Panoramic Video Depth Estimation via Geometry-Aware Spatiotemporal Adaptation"
_ICML.cc/2026/Conference — ICML 2026 regular_

### Official Review · Reviewer_bsyY · 2026-02-23

**Soundness:** 3
**Presentation:** 3
**Significance:** 2
**Originality:** 3
**Overall Recommendation:** 4
**Confidence:** 4

**Summary:**

This paper presents PVDepth, an end-to-end generative framework designed for panoramic video depth estimation. It addresses the lack of large-scale training data and the geometric distortions in Equirectangular Projection (ERP). Contributions contain: a large-scale synthetic RGB-D panoramic video dataset, PSNI module, and CRTM module. Extensive experiments show that PVDepth achieves better performance in geometric accuracy and temporal consistency compared to existing methods. Each module is carefully ablated with experiments.

**Compliance With Llm Reviewing Policy:**

Affirmed.

**Final Justification:**

The rebuttal decently replies to my concerns and I would like to keep my positive score.

**Key Questions For Authors:**

Overall, I think this is a good paper but the incremental property hinds me from giving a higher score. I will also check opinions from other reviewers during rebuttal to give a fair evaluation.

**Limitations:**

yes

**Strengths And Weaknesses:**

- Pros

    1. This submission is technically sound. Instead of proposing a completely new architecture, the authors adapt state-of-the-art models (like DepthCrafter) to the panoramic domain. Two modules (PSNI and CRTM) are carefully designed to solve specific issues with ablation studies to demonstrate their effectiveness.

    2. By introducing PanoCARLA, the authors fill a gap in the community, as most existing panoramic datasets are limited to static or indoor scenes.

    3. Qualitative results on in-the-wild videos from the Web360 dataset demonstrate that the model, despite being trained on synthetic data, generalizes well to real-world scenarios.

- Cons

    1. The scoop of this work is limited to panoramic video depth estimation, which might lead to less impactful to other fields. The specific architectural adaptations (PSNI and CRTM) are tightly coupled with ERP geometry.

    2. From the idea level, the technique is not that novel. The paper brings the idea from SpND for PSNI and CRTM is merely a feature enhance module with projection.

   3. The contribution is more of a "systematic engineering success": integrating existing geometric intuitions into a modern diffusion framework rather than a fundamental shift in framework.

---

> ### Author Rebuttal · Authors · 2026-03-30
>
> We sincerely thank Reviewer bsyY for the constructive feedback. We appreciate the Reviewer's recognition of the technical soundness of our manuscript, the value of PanoCARLA, and the promising real-world generalization of our method.
>
> Below, we address the two concerns: broader impact and novelty.
>
> **1. On the broader impact of PVDepth beyond panoramic video depth estimation (C1)**
>
> PVDepth is indeed developed for panoramic video depth estimation, and PSNI and CRTM are specifically designed for ERP geometry. However, we believe the contribution of this work is not limited to this task alone.
>
> First, beyond the proposed model, we introduce PanoCARLA, which provides synchronized panoramic RGB-D videos together with per-frame camera poses. As such, it can support future research not only in panoramic video depth estimation, but also in other 3D-aware panoramic video understanding tasks and geometry-aware panoramic video generation tasks.
>
> Second, our contribution is also at the level of problem formulation. Beyond the widely recognized issue of ERP spatial distortion, we explicitly identify **ERP-induced non-linear temporal dynamics** as another key challenge in panoramic videos. To the best of our knowledge, this challenge has not been explicitly formulated in prior panoramic video works. Our CRTM module is designed specifically to address this challenge, while PSNI addresses the complementary issue of non-uniform spatial information density. Since these challenges arise from ERP video geometry itself rather than from depth estimation alone, we believe the underlying modeling insights and geometry-aware adaptations may also be informative for other panoramic video tasks.
>
> Furthermore, since our framework is built on a diffusion-based backbone,  the proposed geometry-aware adaptations may also be relevant to panoramic video generation models operating in ERP space. The effectiveness of both designs, as well as their complementarity, is supported by the ablation studies.
>
> We will revise the manuscript to clarify this broader perspective more explicitly.
>
> **2. On the novelty of the work (C2 & C3 & Q1)**
>
> We agree that PVDepth is not a fundamentally new generic depth framework. PVDepth's main contribution is **to close the loop for panoramic video depth estimation as a research problem**. To the best of our knowledge, this is **the first work** to do so by: (1) introducing a large-scale panoramic RGB-D video dataset and establishing a systematic benchmark for evaluating existing methods, and (2) establishing an end-to-end framework that adapts perspective video depth models to panoramic video depth estimation.
>
> More importantly, we explore both cube-based and ERP-based strategies for adapting perspective video depth models to the panoramic domain, and show that direct adaptation remains suboptimal due to the domain gap between perspective and panoramic videos. Based on this observation, we further analyze the source of this domain gap and explicitly formulate two key challenges in ERP videos: **non-uniform spatial information density** and **non-linear temporal dynamics**. We then design PSNI and CRTM to address these two challenges respectively. Their effectiveness and complementarity are consistently supported by the detailed ablation studies. In this sense, the contribution is not merely the addition of two lightweight modules, but the combination of explicit problem analysis, targeted geometry-aware design, and systematic empirical validation.
>
> Regarding module-level novelty, **PSNI** is inspired by prior geometric intuition, but it is not a direct reuse of SpND. Instead of introducing an auxiliary geometric branch, we incorporate spherical priors directly into the noise initialization of the diffusion model, and further design a progressive adaptation strategy to bridge pretrained planar priors and ERP geometry without introducing additional learnable parameters or notable extra computation. Likewise, **CRTM** is not intended as a generic feature enhancement block. It is designed to address ERP-induced **non-linear temporal dynamics** by performing temporal rectification in the cube feature domain and feeding the rectified features back to the ERP branch in a stable manner. Although simple, it brings clear gains in ablation studies, and further improves performance when combined with PSNI.
>
> We will revise the manuscript to more clearly present our contribution.

---

> > ### Author Rebuttal · Reviewer_bsyY · 2026-04-02
> >
> > Overall, my concerns are fully resolved. I would like to keep my positive score.

---

> > > ### Author Response · Authors · 2026-04-02
> > >
> > > Thank you for your positive assessment and for confirming that all concerns have been fully addressed.
> > >
> > > We truly appreciate your thoughtful feedback and support throughout the review process.

---

### Official Review · Reviewer_JqW8 · 2026-03-12

**Soundness:** 3
**Presentation:** 3
**Significance:** 3
**Originality:** 2
**Overall Recommendation:** 3
**Confidence:** 4

**Summary:**

This paper presents an end-to-end framework for panoramic video depth estimation that addresses challenges introduced by equirectangular projection, including spatial geometric distortions and temporally non-linear motion patterns. To address these issues, the authors propose two modules: Progressive Sphere-aware Noise Initialization, which adapts the diffusion noise initialization to spherical geometry, and Cube-rectified Temporal Modeling, which performs temporal modeling in cube-map space to mitigate ERP-induced temporal distortions. In addition, the paper introduces PanoCARLA, a large-scale synthetic panoramic RGB-D video dataset generated in the CARLA simulator. The dataset contains dynamic outdoor driving scenes with natural 6-DoF camera trajectories generated via CARLA’s autonomous navigation policy. Experiments show that PVDepth outperforms existing panoramic and perspective-based video depth estimation methods on this dataset, along with qualitative results on real-world panoramic videos.

**Compliance With Llm Reviewing Policy:**

Affirmed.

**Final Justification:**

As I mentioned, the authors' rebuttal is not enough to address all of my concerns. So, I want to keep my rating.

**Key Questions For Authors:**

1. In Table 1, DepthAnyVideoZS performs better than DepthCrafterZS under the first condition (fps02_len50). Is there a reason why a fine-tuned DepthAnyVideo baseline was not included in the comparison?

2. In Appendix F, both the camera poses and the scale alignment used for evaluation are derived from ViPE, which may introduce a circular dependency. Would PVDepth still outperform other methods if an independent pose estimator (e.g. COLMAP or a SLAM) were used?

3. Since the method expands ERP frames into six cube faces during the denoising process, could the authors report the actual GPU memory usage and FLOPs in addition to latency?

4. The model is trained at 640×320 but evaluated at 1024×512. How sensitive is the method to this train–test resolution mismatch?

**Limitations:**

Yes

**Strengths And Weaknesses:**

- Strengths


Significance and Dataset Contribution : Panoramic video depth estimation is an important problem for applications such as VR, robot navigation, and world modeling, yet relatively few prior works address it in an end-to-end manner. In this context, the introduction of PanoCARLA is a valuable contribution to the community. The planned release of both code and data further increases the potential impact of this work.

Dataset Construction and Design : Unlike simple trajectory interpolation approaches, PanoCARLA generates natural 6-DoF camera trajectories using CARLA’s autonomous navigation policy. In addition, while existing panoramic RGB-D datasets are largely limited to static indoor scenes, PanoCARLA contains dynamic outdoor environments, making it a meaningful addition to existing benchmarks.

Technical Soundness and Module Complementarity : The design choice of addressing ERP-specific challenges with two dedicated modules is clearly motivated. PSNI targets spatial geometric inconsistencies, while CRTM models temporally non-linear dynamics. Their complementary roles are supported by the ablation study in Table 2. Furthermore, PSNI is computationally efficient since it modifies the noise distribution without introducing additional learnable parameters.


- Weaknesses


Evaluation Limitations and Real-world Validation : Most quantitative evaluations are conducted on the synthetic PanoCARLA dataset. Although the paper includes experiments on real-world videos, these evaluations do not use ground-truth depth and rely on ViPE-derived poses and scale alignment, which introduces a potential circular dependency. As a result, the reliability of the real-world comparison appears somewhat limited.

Dataset Domain Bias : PanoCARLA mainly contains outdoor driving scenes generated in CARLA and does not include indoor environments. While the paper provides qualitative indoor examples, the lack of quantitative evaluation makes it difficult to assess the model’s generalization to different scene types.

Efficiency and Practical Applicability : The reported inference latency (~482 ms per frame) appears relatively high for real-time applications such as VR or robotic navigation, which may limit practical deployment without further optimization.

Limited Methodological Novelty : The proposed approach largely adapts existing techniques such as diffusion-based depth estimation and cube-map projections to the panoramic video setting. While the integration is reasonable and well motivated, the overall methodological novelty appears somewhat limited.

---

> ### Author Rebuttal · Authors · 2026-03-31
>
> We sincerely thank Reviewer JqW8 for the careful reading and thoughtful comments.
>
> **Due to the rebuttal length limit, all tables are provided in the anonymous link:** [LINK](https://anonymous.4open.science/r/supplementary_materials-2A40/XXXX-2_materials.pdf). We kindly refer Reviewer to it for details, and will elaborate in the revised manuscript.
>
> **1. Real-world validation (W1 & Q2)**
>
> We agree that Appendix F has limitations because no public real-world panoramic video depth dataset is currently available. Such data are difficult to collect because existing panoramic cameras do not support synchronized RGB-D capture. We therefore use reprojection consistency only as supplementary validation.
>
> Regarding the potential circular dependency in prior evaluation, we further tested a **MapAnything**[1]-based protocol. Since it does not support ERP input, for each ERP video we project a front-view cubemap sequence, estimate depth&poses with MapAnything, then project depth back to ERP along with its valid-region mask. We use above depth&pose for alignment and reprojection. **Table 1 in anonymous link** shows that PVDepth still performs best.
>
> *[1] Mapanything:Universal feed-forward metric 3d reconstruction. arXiv2025*
>
> **2. Dataset domain bias (W2)**
>
> PanoCARLA is indeed outdoor-biased because CARLA only provides outdoor scenes. Our goal is to establish a benchmark and verify methods in a controlled setting.
>
> To the best of our knowledge, there is currently no publicly available indoor RGB-D panoramic video dataset for quantitative evaluation, so we only provide qualitative results. As shown in Fig. 13 of manuscript, PVDepth can generate plausible qualitative results and coarse near–far structure on indoor videos, but we do not over-interpret this as strong evidence of indoor generalization.
>
> We will clarify this limitation more explicitly in the revision.
>
> **3. Efficiency and practical applicability (W3)**
>
> We agree that the current inference latency is insufficient for real-time applications, as also discussed in Sec. 5.4. The main bottleneck is iterative diffusion sampling.
>
> Still, PVDepth (0.482s) is substantially faster than post-optimization-based methods such as ViPE (0.5s-3s) and Matrix-3D (20s), highlighting the significance of end-to-end panoramic video depth models.
>
> In practice, this limitation may be mitigated by diffusion acceleration techniques (e.g., fewer-step inference and distillation[2]) or by operating at a lower frame rate rather than at every frame.
>
> *[2] Marigold:Affordable adaptation of diffusion-based image generators for image analysis, TPAMI2025*
>
> **4. Fine-tuned DepthAnyVideo (DAV) baseline (Q1)**
>
> We did not include a fine-tuned DAV since its official release only provides inference code, and faithfully reproducing its training pipeline is challenging due to its non-trivial components beyond vanilla SVD: Flow matching, RoPE encoding, and especially an additional interpolation UNet. These implementation details are not fully released.
>
> Following Reviewer's suggestion, we attempted a best-effort DAV fine-tuning reproduction based on our understanding. However, the current results are unstable and significantly underperform expectations, suggesting that our reproduction is not yet reliable. To avoid reporting potentially misleading results, we choose not to include them at this stage.
>
> We will continue refining the training setup and provide results once they are reliable.
>
> **5. GPU memory and FLOPs (Q3)**
>
> Following Reviewer's suggestion, we additionally profiled GPU memory and FLOPs, as reported in **Table 2 of anonymous link**. Compared with DepthCrafter, PVDepth uses nearly identical GPU memory but higher FLOPs and latency due to ERP--Cube projection and auxiliary temporal modeling, while achieving clear accuracy gains.
>
> **6. Train–test resolution mismatch (Q4)**
>
> Due to computational constraints, using different resolutions for training&inference is not uncommon in diffusion models; for example, DepthCrafter is trained at 640×320 and also reports experiments at 1024×576. Following Reviewer's suggestion, we evaluated models under multiple test resolutions, as reported in **Table 3 of anonymous link**.
>
> We observe broadly consistent trends across DepthCrafter (ZS), DepthCrafter (FT), and PVDepth: performance improves substantially from low to medium/high resolution and then becomes relatively stable. For PVDepth, although training is conducted at 640×320, testing at higher resolutions remains beneficial, improving AbsRel while maintaining comparable $\delta_1$.
>
> Similar resolution-dependent behavior has also been observed in prior work, such as RA-Depth[3].
>
> *[3] Ra-depth:Resolution adaptive self-supervised monocular depth estimation, ECCV2022*
>
> **7. Limited methodological novelty (W4)**
>
> This concern is closely related to **Reviewer bsyY's comment on novelty**. Due to space limits, we kindly refer Reviewer to our response to Reviewer bsyY, where this point is addressed in detail.

---

> > ### Author Rebuttal · Reviewer_JqW8 · 2026-04-03
> >
> > First of all, thanks for the authors' effort. The authors have partially addressed my concerns and I maintain my score.
> > I have still issues on the novelty of the proposed method, DAV fine-tuning and GPU usage.

---

> > > ### Author Response · Authors · 2026-04-04
> > >
> > > We thank Reviewer JqW8 for the follow-up feedback.
> > >
> > > Due to the rebuttal space limit, **all tables are in our prior anonymous link:** [LINK](https://anonymous.4open.science/r/supplementary_materials-2A40/XXXX-2_materials.pdf). We kindly refer Reviewer to it for details.
> > >
> > > **1. Fine-tuned DepthAnyVideo (DAV)**
> > >
> > > Following Reviewer's suggestion, we completed a best-effort fine-tuning reproduction of DAV and present results in **Table 4 of anonymous link**. It can be observed that, under the same 6-frame training setting, DAV fine-tuning is unstable and even degrades below its zero-shot version. **This also explains why we did not report the results in the first rebuttal: we did not want to present a potentially misleading result before understanding this phenomenon better**.
> > >
> > > By inspecting inference results, we observed depth flickering across different keyframe intervals in the "6-frame DAV model". This phenomenon may stem from its two-stage design: DAV first estimates keyframe depths, then uses an interpolation UNet to estimate intermediate depths. The interpolation UNet appears harder to stabilize under short training clips. In Table 4, training DAV with 16-frame clips substantially improves its performance (although needing more computational resources for training), confirming DAV's sensitivity to training clip length.
> > >
> > > In addition, compared with our PVDepth, DAV appears more sensitive to FPS, with larger variation across FPS settings. A plausible reason is DAV's keyframe-first inference: for low-FPS clips, additional temporal sparsification between keyframes further weakens inter-frame correlation.
> > >
> > > For fairness, we also trained a 16-frame version of PVDepth. PVDepth also improves with longer training clips and achieves better overall performance than DAV.
> > >
> > > **2. GPU usage**
> > >
> > > Since the follow-up comment mentions GPU usage without specifying the exact issue, we clarify the most likely point of confusion: why GPU memory did not increase along with FLOPs in CRTM.
> > >
> > > As reported in **Table 2 of anonymous link**, for 110-frame clips at 1024 × 512 resolution, compared with the baseline DepthCrafter, total FLOPs increase from 1.986 PFLOPs to 2.553 PFLOPs, while GPU memory only increases from 30.6 G to 30.7 G.
> > >
> > > The main reason is that CRTM does **not** maintain a persistent cube-domain feature throughout the UNet. Instead, each temporal block only temporarily performs `ERP -> Cube -> temporal modeling -> Cube -> ERP -> fusion`, after which the intermediate cube features are released. Thus, the extra cost mainly appears in computation rather than long-lived memory. Furthermore, the ERP and Cube temporal branch are executed sequentially, which further explains why latency increases while GPU memory changes little.
> > >
> > > Also, the Cube branch does not inflate spatial tokens by 6× as one might expect. For an ERP latent of width W, the ERP branch has $(W/2) \times W = \frac{1}{2}W^2$ spatial tokens, while six cube faces together contain $6(W/4)^2 = \frac{3}{8}W^2$, i.e., only about 75% of the ERP spatial tokens, not a 6× expansion.
> > >
> > > **3. Novelty**
> > >
> > > Due to the rebuttal space limit, our first response referred Reviewer to our reply to Reviewer bsyY. We further clarify it here.
> > >
> > > PVDepth is not a fundamentally new generic depth framework. Instead, our intended contribution is different: **to close the loop for panoramic video depth estimation** by combining:
> > >
> > > - a large-scale panoramic RGB-D video dataset and benchmark.
> > > - explicit formulation of two ERP-specific challenges — non-uniform information density and non-linear temporal dynamics.
> > > - targeted geometry-aware adaptation rather than purely engineering trial-and-error.
> > >
> > > **At the methodological level**, our contribution is not simply adapting existing diffusion-based depth estimation and cubemap projection to panoramic video. Starting from a strong perspective prior, we explored direct adaptation strategies and found them still suboptimal. Through domain gap analysis, we explicitly formulate the above two ERP spatiotemporal challenges; to our knowledge, the non-linear temporal dynamics has not been explicitly formulated in prior panoramic video works. We then design PSNI and CRTM to address these two challenges, respectively.
> > >
> > > **At the module level**, CRTM is built on the distortion-free property of cubemap representation, but it is specifically motivated by the challenge of ERP-induced non-linear temporal dynamics. This problem-driven design lets us add a lightweight cube temporal branch only in the parameter-efficient temporal layers, rather than a full cubemap-based UNet branch. Likewise, PSNI injects spherical priors directly into the diffusion process instead of adding an extra geometric branch. Although lightweight, both modules bring clear complementary gains in our ablation studies, closing the loop between our problem formulation and corresponding adaptation.
> > >
> > > We will revise the manuscript to make this contribution framing clearer.

---

### Official Review · Reviewer_9vaK · 2026-03-12

**Soundness:** 4
**Presentation:** 4
**Significance:** 3
**Originality:** 3
**Overall Recommendation:** 5
**Confidence:** 4

**Summary:**

The paper tackles the problem of Panorama Video Depth estimation. They identify the lack of training data at scale and the artifacts introduced by the equirectangular projection as key challenges towards solving this problem. In order to address said problems they make three contributions:

1. They propose a large scale training dataset rendered using the Carla simulator. They take special care to avoid persistent occlusion (arising from the car being always in view) by rendering cubemaps using an invisible rig and also deal with inconsistent exposure induced seam artifacts in the stitched equirects by using >90 FoV cube faces and feather blending.
2. Following prior work they use equirect distortion aware noise (instead of iid noise) for denoising. Differently from prior work, they propose a weighted sum of the iid and equirect distortion aware noise and further propose a curriculum for adapting to the new noise schedule.
3. They propose applying temporal attention in the projected cubemap space

They evaluate their method against recent image and video depth baselines in Table 1 and achieve SOTA results. Each of the contributions is convincingly ablated.

**Compliance With Llm Reviewing Policy:**

Affirmed.

**Final Justification:**

The rebuttal addressed the minor concerns I had. I choose to maintain my score.

**Key Questions For Authors:**

Minor clarification questions:

Q1: Could the authors provide more details on how the noise is sampled on a 3D grid (e.g. what is the grid size) and how it mapped to ERP (e.g. what is the interpolation method used)

Q2: Could the authors expand on what they mean by "This stems from its semantic masking strategy: by optimizing scale solely on the static background, it introduces bias when applying this global scale to dynamic foreground objects" in section 5.2. Why is optimizing scale on the static background not sufficient?

**Limitations:**

Yes

**Strengths And Weaknesses:**

Soundness: The contributions are sound and their efficacy is clearly demonstrated by the SOTA empirical results and the ablations.

Presentation: The work is polished and easy to follow and adequately discusses and compares against relevant prior work.

Significance: Pano video depth remains an important and unsolved problem suffering from a lack of the training data and insufficient research into modeling techniques. This work is a useful contribution in direction.

Originality: There are several novel contributions such as the PanoCarla dataset, the proposed noise schedule and the architectural changes.

---

> ### Author Rebuttal · Authors · 2026-03-30
>
> We sincerely thank Reviewer 9vaK for the positive and encouraging feedback. We appreciate the Reviewer's recognition of the significance of our work and the soundness of our dataset, method design, and experimental results.
>
> Below, we address the two clarification questions in detail.
>
> **Q1: More details on the 3D grid noise and its mapping to ERP**
>
> Our goal is to construct a sphere-aware noise initialization that is consistent with the underlying spherical geometry of Equirectangular Projection (ERP). To make this practical, we first sample a Gaussian noise field on a 3D grid and then project it onto the ERP latent through a fixed spherical sampling operator.
>
> Specifically, for an ERP latent of size $ H_l \times W_l $, we use a 3D noise grid of size $ D \times D \times D $, where $ D = W_l / 4 $, following the usual ERP-to-cubemap resolution correspondence. For example, an ERP input of $ 512 \times 1024 $ corresponds to a latent of $ 64 \times 128 $, for which we use a $ 32 \times 32 \times 32 $ 3D grid.
>
> The projection from the 3D noise grid to ERP is implemented as follows. For each ERP latent pixel, we compute its longitude and latitude, and convert it into a unit 3D direction:
>
> $(x, y, z) = (\cos\theta \cos\phi, \cos\theta \sin\phi, \sin\theta)$,
>
> where $ \theta \in [-\pi/2, \pi/2] $ is latitude and $ \phi \in [-\pi, \pi] $ is longitude. Collecting these directions over all ERP pixels gives a precomputed ERP-to-3D coordinate map of shape $ H_l \times W_l \times 3 $. This map specifies the 3D sampling location in the 3D grid for each ERP latent pixel. Since this location generally falls between discrete grid points, the final ERP noise value is obtained by **trilinear interpolation**.
>
> An important practical detail is that, for a fixed ERP latent resolution, this ERP-to-3D coordinate map is deterministic and depends only on the target resolution. Therefore, it needs to be computed only once and can be cached and reused for all samples during training and inference, so the additional computational overhead is negligible in practice.
>
> Through the above process, the resulting ERP noise is consistent with the underlying spherical geometry, and its information density is naturally aligned with the sphere-to-ERP projection.
>
> We will incorporate these implementation details into the revised manuscript for clarity.
>
> **Q2: Why might optimizing scale on the static background be insufficient?**
>
> In our manuscript, this sentence was intended to provide a possible interpretation of why ViPE improves $\delta_1$ but degrades AbsRel relative to its UniK3D baseline.
>
> Specifically, ViPE estimates masks for dynamic objects and performs geometric optimization primarily on the masked static background. This design is reasonable for stabilizing camera pose and background geometry. However, in dynamic scenes, foreground objects may exhibit different depth errors from the background due to independent motion and occlusion. Therefore, a single global affine correction estimated from static-background regions may not transfer equally well to dynamic foreground objects. As a result, the optimization can improve overall geometric consistency while still leaving misalignment on moving objects, which may partly explain the discrepancy we observed between $\delta_1$ and AbsRel.
>
> To make this point more precise, we will revise the manuscript to clarify that this is an interpretation based on ViPE's masking-based optimization strategy rather than a definitive causal conclusion.
>
> We will revise the original description as follows:
>
> _This may be related to ViPE's semantic masking strategy: its global affine depth alignment is estimated primarily from static-background regions, so the resulting correction may not transfer equally well to moving foreground objects in dynamic scenes._
>
> We will incorporate the above clarifications into the revised manuscript to improve precision and readability.

---

> > ### Author Rebuttal · Reviewer_9vaK · 2026-04-02
> >
> > The authors have addressed my concerns and I maintain my recommendation for acceptance.

---

> > > ### Author Response · Authors · 2026-04-04
> > >
> > > Thank you very much for your consistent support throughout the review process.
> > >
> > > We truly appreciate your careful feedback and are glad that our responses addressed your concerns.

---

### Decision · Program_Chairs · 2026-04-30

**Decision:**

Accept (regular)

**Comment:**

This paper receives mixed reviews. Reviewers acknowledge that the newly introduced dataset, consisting of large-scale panoramic video data created using the CARLA simulator, is valuable to the community. To address the issues of spatial geometric distortions and temporally non-linear motion patterns, two modules are introduced in the paper. Reviewers find the proposed modules well motivated and well validated in the ablation studies. However, reviewers bsyY and JqW8 find that the technical novelty is limited, as the PSNI module is adapted from SpND and CRTM for feature rectification. The authors provided a detailed response and justified that the adaptation is not trivial.

The AC checked the reviewers’ comments, the authors’ rebuttal, the paper, and the supplementary material. The proposed new dataset is valuable. While it does not cover indoor scenes, its extension of CARLA to PanoCARLA benefits the field. The new pipeline is also reasonable.

The AC would therefore recommend acceptance.